# Modeling the Dynamics of T-Cell Development in the Thymus

**DOI:** 10.3390/e23040437

**Published:** 2021-04-08

**Authors:** Philippe A. Robert, Heike Kunze-Schumacher, Victor Greiff, Andreas Krueger

**Affiliations:** 1Department of Immunology, University of Oslo, 0372 Oslo, Norway; victor.greiff@medisin.uio.no; 2Institute for Molecular Medicine, Goethe University, 60590 Frankfurt, Germany; Heike.Kunze-Schumacher@kgu.de

**Keywords:** thymic selection, T-cell development, T-cell receptor (TCR), mathematical modeling, multi-scale models, complex systems, ordinary differential equations (ODE), agent-based models

## Abstract

The thymus hosts the development of a specific type of adaptive immune cells called T cells. T cells orchestrate the adaptive immune response through recognition of antigen by the highly variable T-cell receptor (TCR). T-cell development is a tightly coordinated process comprising lineage commitment, somatic recombination of Tcr gene loci and selection for functional, but non-self-reactive TCRs, all interspersed with massive proliferation and cell death. Thus, the thymus produces a pool of T cells throughout life capable of responding to virtually any exogenous attack while preserving the body through self-tolerance. The thymus has been of considerable interest to both immunologists and theoretical biologists due to its multi-scale quantitative properties, bridging molecular binding, population dynamics and polyclonal repertoire specificity. Here, we review experimental strategies aimed at revealing quantitative and dynamic properties of T-cell development and how they have been implemented in mathematical modeling strategies that were reported to help understand the flexible dynamics of the highly dividing and dying thymic cell populations. Furthermore, we summarize the current challenges to estimating *in vivo* cellular dynamics and to reaching a next-generation multi-scale picture of T-cell development.

## 1. Introduction

The thymus is a unique environment. It is the site of T-cell development. At steady state, it is dependent on continual colonization by a very low number of bone-marrow-derived progenitor cells (for review see [1]). In experimental systems in which influx of T-lineage competent progenitors is perturbed, T-cell development may be sustained for extended periods of time, by self-replication of thymocytes that already reside in the thymus [2,3]. Thymic size and output are dynamic. The thymus gradually involutes with age, and can transiently shrink up to 90% under stress, pregnancy or infection [4]. Surface markers allowed delineation of many sub-populations of developing T cells (the thymocytes), corresponding to key steps of development and selection. Their dynamics have been extensively measured *in vivo* following organ reconstitution after irradiation, injection of labeled progenitors, thymic grafts, or *in vivo* labeling. Furthermore, the development of thymocytes involves the decision to differentiate into several downstream populations either carrying an αβTCR, as CD8 T cells, Foxp3${}^{-}$ CD4 T cells, Foxp3${}^{+}$ regulatory T cells, but also as unconventional T cells carrying either αβ or γδ TCRs [5]. This complexity has sparked the design of population-based mathematical models to understand the dynamical properties of T-cell development and differentiation in the thymus, and predicted the existence of feedback regulation yet to be verified experimentally. Interestingly, despite the large amount of available data, it is still very complicated to identify the death and proliferative behavior of thymocytes, in particular the duration of their cell cycle. This knowledge gap limits our understanding of the quantitative regulation controlling T-cell development, and mathematical models are well suited to infer such quantitative parameters hidden inside complex experimental datasets.

The thymus is also known for its substantial quality control of thymocytes. After they have somatically rearranged their TCR loci by V(D)J recombination, it has been estimated that more than 90% of thymocytes die through a process called thymic selection (see Section 4). During selection, developing thymocytes continually probe antigen-presenting cells (APCs) for interactions between (peptide-)antigen presented on major histocompatibility complex proteins (pMHC) and their TCR. High-affinity interactions or failure to interact with sufficient affinity result in death by neglect or clonal deletion, respectively, ideally resulting in the formation of T cells with a broad, yet non-autoreactive, TCR repertoire.

Here, we complement previous reviews on thymic selection theories [6] and quantification of T-cell development [7] by providing an updated view of mathematical modeling approaches of the dynamics of T-cell development in the thymus. To this end, we focus on the description of experimental approaches that are suited to provide quantitative information. We describe mathematical models derived from such experiments and discuss how advances in experimental data foster refined mathematical models and vice versa. We deliberately omit mathematical models studying the quantitative impact of positive and negative selection onto the produced repertoire, pathogen escape or MHC recognition, which are already comprehensively described in [6] and were not extensively revisited since then. Instead, we focus on the complexity of inferring *in vivo* T-cell development properties from sometimes indirect experimental settings. Every model relies on assumptions and simplifications needed to match the complexity of the available experimental dataset. We discuss how experimental and model design limitations may be overcome in future studies.

After describing population dynamic models, models to infer cell-cycle speed in the thymus *in vivo*, and estimation of cell death through the selection steps, we highlight pioneering models that link thymocyte motility and signaling to cell fate and population dynamics. We discuss how next-generation models may be formulated in the context of novel high-throughput TCR sequencing technologies.

## 2. A Journey through Population Models of T-Cell Development

The main steps of T-cell development in the thymus are depicted in Figure 1A and described in Box 1. The earliest T-cell progenitors in the thymus form a subset of the so-called DN1 (double-negative, lacking the expression of CD4 and CD8) cells and are also referred to as Early T-lineage Progenitors (ETP) [8,9]. They arise from bone-marrow-derived cells transiting via the blood. It has been estimated that only a few cells can enter the murine thymus, with a model of ‘gated entry’ where one cell can fill one out of 160 available niches, a number derived from multicongenic barcoding of progenitors followed by mathematical simulation [10,11]. The mechanisms underlying gated entry remain elusive. Periodic alterations in levels of chemoattractants as well as, yet to be molecularly defined, gated release of progenitors from the bone marrow have been proposed [12,13]. Once inside the thymus, an ETP undergoes multiple divisions before sequentially becoming DN2, DN3 and DN4 based on expression of the surface markers CD25 and CD44 [14,15] (Figure 1B).

Box 1Trajectory of murine intrathymic T-cell development.Thymocytes can be broadly characterized based on their surface expression of the co-receptors CD4 and CD8. The most immature thymocytes are negative for both co-receptors and are hence referred to as double-negative (DN). They give rise to CD4 and CD8 double-positive (DP) thymocytes followed by loss of one of the co-receptors to form CD4 or CD8 single-positive (SP) mature thymocytes, which egress from the thymus after final maturation. Upon entry into the thymus, bone-marrow-derived progenitors give rise to early T-lineage progenitors (ETPs), phenotypically characterized as CD44${}^{hi}$CD117${}^{hi}$CD25${}^{-}$. ETPs constitute a subpopulation of the heterogeneous DN1 (CD44${}^{hi}$CD25${}^{-}$) population. Acquisition of CD25 marks the next developmental DN2 stage. At this stage, T-lineage commitment is completed and pre-commitment and post-commitment DN2 thymocytes are referred to as DN2a and DN2b, respectively. DN2b cells express somewhat lower levels of CD117, which progressively decline towards the CD44${}^{-}$CD25${}^{+}$ DN3 stage. V(D)J recombination of Tcrb, Tcrg and Tcrd loci commences at the DN2b stage and continues in a subset of small DN3 cells, termed DN3a (CD44${}^{-}$CD25${}^{hi}$CD27${}^{-}$CD28${}^{-}$). Upon successful V(D)J recombination, DN3a cells give rise to either γδT cells or large DN3b cells (CD44${}^{-}$CD25${}^{int}$CD27${}^{+}$CD28${}^{+}$) in a process called β-selection. Progressive loss of CD25 marks the DN4 compartment, which in turn gives rise to pre-selection DP thymocytes (CD4${}^{+}$CD8${}^{+}$TCR$\alpha \beta^{low/neg}$CD69${}^{-}$CD5${}^{-}$) via an immature CD4${}^{-}$CD8${}^{+}$TCR$\alpha \beta^{-}$ (ISP) intermediate. At the pre-selection DP stage rearrangement of the Tcra locus occurs followed by the initiation of selection. Positively selected DP thymocytes up-regulate the αβTCR and acquire expression of CD69 and CD5. Loss of one co-receptor marks generation of CD4 and CD8 SP thymocytes, whose maturation is further characterized by loss of CD69 and CD24 as well as acquisition of CD62L and MHC-I.

### 2.1. Early Steps of T-Cell Development

The dynamics of DN1 to DN4 cells have been monitored by injection of congenic bone-marrow-derived progenitors [18]. Injected cells remained at the DN1 stage for 10–12 days while transition through the DN2 population was short as DN3 cells appeared after as early as 11 days, and DN4 cells after day 14–15. A mathematical model from Manesso and colleagues [19] used this dataset to compare different proliferation model structures for the DN1 population. The types of equations are depicted in Figure 2A and the model structure in Figure 2B. The best model fit predicts that cells would remain in DN1 for up to 11 divisions before transitioning to DN2s, spending on average 1 day per cycle. Interestingly, no other hypotheses, in which cells would leave the DN1 stage after fewer divisions, or with more distributed probabilities to leave DN1 at earlier divisions, could explain the data well, revealing a synchronization of the cells to leave after a certain number of divisions (or time). This prediction was further experimentally supported by showing a higher differentiation potential of late DN1s [19] as well as progressive transcriptional changes allowing the definition of a developmental trajectory within ETPs [20].

Although identified parameters for the DN1 population and the synchronization statement were robust to the Porritt dataset [18], the inferred residence or cycling times for the DN3 and DN4 populations are not identifiable from this dataset, meaning the exact same curves can be reproduced with different cycling speed of these populations due to compensation between parameters. This means additional experimental constraints would be required to also fix the DN3 and DN4 dynamical parameters, and likely comes from the fact that the dataset could only monitor the frequencies of labeled donor cells rather than absolute numbers, possibly due to a high variation of progenitor engraftment among transplanted mice. Altogether, the model was useful to uncover the synchronized behavior of DN1s and suggests 11 divisions in 11 days for these particular cells. Notably, the study by Porritt and colleagues relied on purification of a limited subset of progenitors, and therefore does not necessarily reflect the physiological composition of thymus seeding progenitors. Some omitted progenitors may in fact display more rapid intrathymic differentiation kinetics [21,22].

Since every 9 to 12 days a new wave of progenitors is initiated [11], it raises the question how thymus size is maintained over time, and in particular, whether cyclic colonization by progenitors would induce detectable fluctuations. The ‘synchronous development hypothesis’ states that the periodic seeding induces such fluctuations, while an opposing hypothesis argues that an asynchronous release of seeders or the existence of size regulation within DN populations could smoothen such fluctuations to undetectable levels. Cai et al. [23] developed a model of DN2-3, DN4 and the subsequent DP compartment without any size regulation and predicted fluctuations to be around 40% amplitude for the DP and total thymocyte populations while DN4 and SP would be quite stable. If this were true, this would mean to expect a high biological variation between different unsynchronized mice. The authors propose a statistical test based on plotting different populations in the same x-y axis, expected to show an ellipse from only one mouse and experimental time-point, if such fluctuations exist. The prediction has not yet been verified. As a replacement for a direct longitudinal analysis of thymocyte numbers, which is not possible, an approximation via ultrasound-based determination of thymus size might be an alternative valid approach.

Bone-marrow-derived thymus seeding progenitors most likely comprise multiple cell types, including IL-7R${}^{+}$ CLPs (common lymphoid progenitor), Flt3${}^{+}$ LMPPs (lymphoid-primed multipotent progenitors) and possibly others, as well as phenotypically ill-defined intermediates [1,21,22,24,25]. For instance, *in vivo*, CLPs displayed a more rapid differentiation into DP thymocytes when compared to LMPPs, suggesting that population heterogeneity of thymus seeding progenitors could contribute to continuous thymic output despite gated entry [7,22].

In general, despite possible variations due to the periodic seeding over weeks and the slow thymic involution over years, most models for thymic populations could fairly consider every population to be at ‘steady state’ during the time of simulation (a few days typically). During the next steps of DN development, the Tcrb locus is genetically recombined and in-frame recombination results first in expression of TCRβ in complex with a surrogate pre-TCRα chain, defining completion of the DN3a stage [26]. Somatic recombination is accompanied by cessation of proliferation and death of cells that fail to productively recombine the Tcrb locus, called β-selection (Figure 1A,B), estimated to kill around 70% of the cells through this checkpoint [27]. Productive recombination of TCRγδ can also happen at this stage and lead to the separate differentiation of γδT cells (Figure 1B). The DN3b and DN4 stages are highly proliferative, and are accompanied by up-regulation of CD8, then both CD4 and CD8 to become ‘immature SP8’ (iSP8) then ‘Double-Positive’ thymocytes (DP), respectively. The latter can be further separated as ‘pre-selection’ DPs and ‘post-selection DPs’ (Figure 1B). Maturation from DN3b to pre-selection DP is a continuous process that comprises massive proliferation followed by recombination of the Tcra locus. Selection is then initiated to probe for formation of a functional TCRαβ complex expressed on the surface. Failure results in death by neglect, which also eliminates cells with αβTCRs with low affinity interactions for pMHC. Successful positive selection is accompanied by expression of activation markers such as CD69. DPs with surface expression of a functional αβTCR are also the first population to be probed for high-affinity pMHC interactions during negative selection resulting in massive cell death [28] (see Section 4).

Please note that those DP thymocytes are defined as “post-selection DPs”, which show phenotypic signs of having received a TCR signal, i.e., such cells have received an initial selecting signal. Nevertheless, they can still be audited for negative selection. It is only partially understood how TCR signaling received through sequential interactions with pMHCs is integrated into apoptosis or differentiation. However, there is evidence that kinetic differences in activation of signaling modules downstream of the TCR as well as differences in their spatial intracellular redistribution contribute to discriminating positively and negatively selecting pMHC ligands [29,30,31,32,33] and it has been proposed that these differences integrate the duration of pMHC-TCR interactions [34]. It remains unknown how single-cell decisions explain the population dynamics of the thymus.

The final step of T-cell development is the choice between becoming a CD4${}^{-}$CD8${}^{+}$ single-positive T cell (future CD8 cytotoxic T cell) or a CD4${}^{+}$CD8${}^{-}$ single-positive T cell, (future conventional CD4 T helper cell (Tconv) or Foxp3${}^{+}$ (Treg cell). Except for Treg-cell precursors, the SP populations are not particularly proliferating, although several studies suggest one or two divisions at post-selection DP or SP stages [35], possibly antigen and MHC-dependent [36] (see Section 3.1). This low-level proliferation can become important when interpreting the parameter values of mathematical models and is noted in (Figure 1A) as a low proliferation arrow.

### 2.2. Estimation of the Flow between Compartments at Steady State Using Larger Models

Apart from the studies from Manesso et al. [19], and the one of Cai et al. [23], analysis of the DN differentiation steps by mathematical modeling has been scarce. A recent transcriptional multi-scale model by Olariu et al. [37] is discussed in Section 5. Comparatively small population sizes render *in vivo* analysis of early developmental stages difficult. Furthermore, well-established *in vitro* differentiation assays may yield unreliable quantitative parameters, as they can be directly influenced by external factors, such as cytokine concentration in culture media. Finally, the original naming of populations into DN1 to DN4 is biologically inconvenient because DN3a cells are more similar to DN2 than DN3b, which in turn are similar to DN4 forming a continuum that is likely to extend to DP cells prior to initiation of Tcra rearrangement. Therefore, one would need to be careful which compartments to simulate and how to associate death and proliferation at the proper stage. The DN2-3a and DN3b-DN4 could possibly be merged as functional compartments, and one would expect a high death rate at the DN3a–DN3b transition. Hence, most other thymic models considered the combined DN stages as one compartment and simulate the major populations of the full thymus, selected according to the biological question of interest.

Inferring the duration of each developmental stage and the flow of cells between them at steady state has been approached both experimentally and mathematically.

Turnover of thymocyte populations has been estimated based on *in vivo* labeling of cells with nucleoside analogues, such as [3H]-thymidine, BrdU and EdU. These labels are incorporated into the cell’s DNA during replication, i.e., they label actively cycling cells. Label incorporation is detected through autoradiography, antibodies, and click chemistry, respectively. Administration of a single pulse allows determination of the frequency of actively cycling cells (see Section 3), whereas continuous labeling allows the determination of turnover within a population by measuring replacement of non-labeled with labeled cells or vice versa. Continuous labeling cannot discriminate between intra-population proliferation and influx of labeled progenitors. Similarly, discrimination between death and outflux of non-labeled progenitors is impossible. Thus, both pulsed and continuous labeling must be complemented with additional assays or mathematical inference to discriminate between these parameters.

Using such sets of experiments, the lifetime of DP thymocytes has been determined to be 3.5 days [38]. Given that most DP cells have a comparatively low rate of proliferation, whereas all DN4 precursors proliferate rapidly, most of the label accumulation can be ascribed to influx. The same study indicated a fraction of only 3% of DP cells becoming SP based on the flow of label to the next generation. A gap in the acquisition of label in SP cells supported the notion that they were largely non-cycling, and their lifetime was estimated to be 12 to 14 days, which may be an overestimate, potentially due to the presence of thymus-resident cells. Analysis of cellular flow through more immature populations was complicated by proliferating populations being interspersed with less proliferating ones [27,39]. These limitations were partially overcome using RAG-deficient and TCR-transgenic strains to interfere with developmental checkpoints [27]. Together these studies revealed population heterogeneity of the DN3 population, consistent with the later identification of DN3a and DN3b subsets [39]. Together, it was proposed that thymocytes undergo approximately 10 divisions between the DN3 to the DP population, and that 70% of DN3 thymocytes die at the β-selection checkpoint [27].

A more recent continuous labeling study showed that pre-selection DP thymocytes successfully transiting to the post-selection DP stage did so within 4 to 5 days [40], and that they display massive caspase activation after 3 days. Using continuous labeling as well, ref. [41] showed that post-selection DPs become fully labeled in 3 to 4 days; naive CD8SP and CD4SP gradually become labeled between day 2 and 8. This shows that the post-selection DP stage is around 3 to 4 days, while the replenishment of CD4 and CD8 might not be synchronous, some cells becoming single-positive more rapidly than others, thus refining the earlier study by Egerton and colleagues [38]. Sinclair et al. [42,43] used a tetracycline inducible Tet murine model, where TCR signaling is blocked by default and developing thymocytes are stuck at the pre-selection DP stage. Treatment with tetracycline rescues T-cell signaling, leading to a synchronized wave of cells from the pre-selection DP stage through positive and negative selections.

In parallel, several mathematical models have been developed to estimate how many cells transit between the populations (Figure 2B). A founding model was published in 1995 [44] for DN, DP, CD4SP and CD8SP populations, where the DN compartment is regulated by logistic growth, and DP and SP populations being regulated by the size of the full thymus. Although no kinetic datasets were available at the time, realistic boundaries for the model parameters were inferred from steady state, from qualitative knowledge and developmental timing known at the time.

As a follow-up, Sawicka et al. [45] have used steady state values from WT mice to identify the flow of cells entering and leaving the DP and SP compartments with single ODEs per population but without size regulation since it was based on steady state. They assumed that SP cells proliferate but not DP thymocytes. Including newer estimations of death by selection from [28], they identified that 35 million cells would enter the DP compartment per day, and give realistic, proliferation and export in each compartment to match the previously estimated residence times and death rates in the thymus. The lack of proliferation in the pre-selection DP compartment likely over-estimates the inflow of cells in the DP compartment, which is probably in the order of a few millions per day since the upstream DN3-DN4 compartment is typically less than 4 million cells (depending on the murine background and age).

A major step for evaluating cell flow rates was the experimental measurement of a developmental wave through the DP and SP populations. The model of Sinclair et al. [42,43] has used the tetracycline-induced developmental wave of cells through post-selection DP and SP stages to infer the flow of cells through CD4 and CD8 differentiation and selection. Their model consists of linear ODEs (Figure 2B), and delineates a 2-step differentiation pathway for CD4 (DP1 and DP2) and a 3-steps pathway for CD8 T cells (DP1 to DP3), which are believed to differentiate later than CD4 T cells from DP thymocytes. T cells with CD8 or CD4 biased TCRs evolve as separate populations with different parameters, and DP1 refers to pre-selection DP. The authors did not assume proliferation at any stage, restricting the main factors to be death, forward differentiation and thymic output. The ratio between death and output at the last stage was inferred by an additional experimental blockade of trafficking using FTY720 treatment [46]. The authors confirmed the robustness of the inferred parameters by bootstrapping. In the model, the larger steady state amount of CD4 SP cells in the thymus compared to CD8 SP cells was not due to a preferential differentiation into CD4 (nor an imbalance in TCR-bias among pre-selection cells), but rather a much larger death rate of CD8-biased T cells during DP stages. The authors discussed a limitation of the inducible Tet experimental system, where T cells show a skewed CD4 vs. CD8 differentiation ratio in comparison to WT mice, likely due to the manipulation of TCR signaling. Although the hypothesis of non-proliferation in post-selection DP stages is experimentally supported, exclusion of limited proliferation in SP [35] and pre-selection DP cells might slightly affect the identified parameters, yet including proliferation would likely create structural correlation between parameters and require additional experimental data to separate proliferation rates from death/export.

### 2.3. Models for Thymus Involution and Shrinkage

The thymus shows an intriguingly dynamic cellularity during life (the number of cells inside each population). First, its size progressively involutes with time, associated with a decrease in both proliferation and survival of the cells [47]. Second, it considerably shrinks following pathophysiological perturbations such as infection, stress, chemotherapy or malnutrition [4]. For instance, *Trypanosoma cruzi* infection induces a slow decay of all populations for 15–20 days and is associated with DP thymocyte death and the unexpected presence of DP cells in the periphery [48]. Pregnancy also induces thymic atrophy for a longer period [49], which could be induced by injection of estradiol in non-pregnant mice. Estradiol-induced atrophy was linked with loss of DN cells and reduced proliferation after β-selection, but did not seem to affect DP cells although Treg-cell development was altered [50]. Thymic atrophy in the context of acute or viral infection such as influenza has gained interest due to recent reports showing the presence of the virus in the thymus [51], either by direct infection due to proximity with the lungs, or imported by migratory APCs coming from the lung [52], which might present foreign antigens as self during selection.

Apart from their pathophysiological relevance, transient perturbations have been employed experimentally to infer dynamical properties of T-cell development in the thymus or to compare mechanistic hypotheses to explain the perturbation.

A first full thymus model built on experimental kinetics has been introduced by Thomas-Vaslin et al. [53]. The authors induced death of proliferating cells and measured the dynamics of thymus shrinkage and recovery, using a conditional suicide gene and injection of an activating compound. The data helped to calibrate a model where DN, early DP (pre-selection DPs) and SP cells can proliferate, while late DPs (post-selection DPs) die by neglect or by negative selection. Interestingly, instead of a single linear ODE per population, they developed a generational model for each proliferating compartment (Figure 2A,B) with a fixed number of divisions (with a fraction of cells exiting before the last division to have smooth average numbers of divisions). From an estimation of 20,000 cells per day entering the DN compartment, they assume that DN cells divide 4 times, during a period of 18 days, while early DPs proliferate 5 to 6 times with high speed (4 to 5 divisions per day). Explaining the experimental rebound requested a very high speed of early DP division in the model. They also estimated that CD4SP and CD8SP would divide between 1 and 2 times and provide an estimation of thymic flow of cells between each compartment including the spleen and lymph nodes together with estimated residence times in each compartment that was consistent with literature.

Newer findings would suggest possible adaptations in the model design. The inflow of 20,000 cells per day entering the DN suggests the DN compartment was referring to DN2-DN3-DN4, as DN1 cells harbor many divisions [19]. The slow proliferation of DN cells with 4.5 divisions in 18 days could be compensated by including death by β-selection, in which case the cells would divide more and faster. Furthermore, separation of the DN compartment into pre- and post-β-selection DNs could allow for higher proliferation of the DN3b-DN4 compartment. In turn, this could result in an increased flow of cells entering the early DP population, therefore requiring more realistic, slower divisions at the early DP stage to get the fast rebound. Finally, the absence of simultaneous proliferation and death, estimated as a single parameter, could be re-interpreted with newer experimental estimates of cell death.

Altogether, the model of Thomas-Vaslin et al. [53] brought substantial contributions to the field. First, it showed that it is possible to explain the dynamics of this strong experimental perturbation with a simple model and without any size regulation nor feedback. Indeed, we have noticed that single linear ODE models typically need to include a logistic growth to get a faster recovery. It is likely that the generational model of Thomas-Vaslin allows for faster reconstitution because cells cannot progress to the next developmental stage until a few divisions are completed whereas linear ODE models have a constant exit rate. Second, the separation of proliferating early DP and highly dying late DP compartments has a realistic model structure and replicated the time-resolved experimental perturbation dataset, suggesting it can be re-used to build more precise models with newer hypotheses such as the one provided by Elfaki et al. [54]. Third, their experimental dataset is valuable to test any new model for T-cell development.

As a different source of atrophy, Moleriu et al. induced thymic atrophy by dexamethasone injection in mice, which triggers cell death, as a surrogate to mimic stress-induced atrophy [55], and used Mehr’s model to identify population dynamical parameters [44]. The dynamics of dexamethasone in the blood are modeled as different possible time-dependent functions. The effect of dexamethasone is modeled as perturbation at the level of proliferation, death, or transfer rates, proportional to the dexamethasone levels. The same dynamics of perturbation applied to all DP and SP populations was not successful in replicating the dynamics, but rather each population needed a perturbation with different strength/dynamics. They also showed that in the model, the proliferation rate and the carrying capacity of the populations were structurally correlated (they compensate each other), meaning that one parameter needs to be fixed arbitrarily, or maybe that a regulation of population sizes is not necessary to explain this dataset. It is unclear whether the atrophy could be explained by a simpler perturbation model using a different differentiation model structure. For instance, in Elfaki et al. [54], atrophy could already be well explained by the dynamic perturbation of only one compartment (increasing death of DP cells). Nevertheless, Moleriu et al. have provided a detailed explanation how far Mehr’s model can be used to infer dynamics of thymic populations.

Recirculation of mature T cells, in particular of Treg cells, is a comparatively novel concept and has not been incorporated in the models described above. Furthermore, development of Treg cells is characterized by dynamic properties distinct from those of Tconv cells, such as their increased rates of proliferation. Elfaki et al. followed influenza-induced thymic atrophy in mice [54], reaching a 90% shrinkage in total cell numbers 7 to 10 days after infection, followed by a very fast recovery of 3–4 days, without prior knowledge of the mechanisms of atrophy. The authors used a RAG1${}^{GFP}$ reporter to distinguish newly generated, RAG${}^{+}$ cells from resident or recirculating cells and asked whether influenza would skew the differentiation of T-cell populations, including Treg cells. By following the dynamics of the main populations during influenza-induced atrophy, they could show that only RAG${}^{+}$ newly generated cells were impacted. The diversity of the Treg TCR repertoire was lower at the peak of atrophy, and the frequencies of Treg populations appeared to be transiently increased. To disentangle the mechanisms by which influenza induces atrophy, they developed a mathematical model, based on the early DP–late DP compartments of Thomas-Vaslin [53]. They adapted the SP populations to include three possible generation pathways for Treg cells, using single ODEs with proliferation and death, and fixing most parameters from literature (Figure 2B). Most parameters for Treg generation are unknown and were fitted. Death, proliferation and output of each SP population were structurally correlated, so the authors fixed their sum (death + output – proliferation) from steady state constraints and experimental residence times. The dynamics of atrophy were completely insensitive to the contribution of death versus output and proliferation provided their sum was constant. A carrying capacity defined by the full thymus size was added to the early DP generational model by changing the number of generations in a smooth manner (with a dynamical output rate for the last two early DP generations, depending on the logistic regulated number of divisions). This regulation was not necessary to explain the atrophy rebound per se, but provided a slightly better fit, thus written ‘optional’ in Figure 2B. The mechanistic impact of influenza did not seem to be direct, as influenza viremia peaks typically much earlier than the peak of atrophy at day 10, suggesting the existence of a downstream factor inducing atrophy, such as glucocorticoids or IFN-γ production by NK or CD8αα cells [54]. Therefore, the authors hypothesized a downstream factor of unknown timing, as a Gaussian perturbation to select population death or differentiation. Interestingly, transiently increased DP death alone could explain well the dynamics of all DP and SP populations, including the observed transient increase of Treg cells as a fraction of the CD4SP compartment. This peak was a dynamical artifact likely due to different lifetimes, where Tconvs decay faster than Foxp3${}^{+}$ populations and the frequency of the latter transiently increases as an overshoot. Modulation of Treg differentiation did not help to explain the data better, but instead, an increased export of all SP thymocytes could improve the fit. This shows the importance of mathematical modeling in understanding the dynamic behavior of populations under perturbations. Consistent with previously defined differentiation trajectories of Tregs [56,57], generation of Treg precursors from CD4SP cells rather than directly from DP precursors provided the best explanation of the data in the study of Elfaki et al. [54], showing that the dynamical perturbation included biological information on Treg ontogeny. It remains an open question, how thymic atrophy decreases Treg TCR diversity and whether this leaves an imprint on the generated repertoire through life. The model showed that the total increased export is minor, meaning that a difference in exported TCR diversity might not have a strong effect on the peripheral repertoire. An agent-based model with cells carrying diverse TCRs could help linking population dynamics to TCR diversity and uncover potential regulatory mechanisms. For instance, reduced Treg diversity could arise from a ‘wrong’ timing of crossing the cortico-medullary junction that is a region with increased antigen presentation. Indeed, modification of thymocyte migration between cortex and medulla does not change the amount of generated Tregs [58,59] but likely impacts the type of encountered antigens. Alternatively, *de novo* Treg formation could occur via different developmental intermediates, which generate Tregs of distinct self-reactivity and functionality [60,61]. Such agent-based model could explain why a change in diversity is unnoticed when it comes to dynamics.

Finally, the natural thymic size involution during the very early stages of development has been modeled in the study by Zaharie et al. using a linear ODE model [62] adapted from Mehr and Moleriu’s models (Figure 2B). Pre-natal and post-birth development are simulated with two different sets of parameters, and thymic involution with age is simulated as an exponentially decreasing proliferation rate of each compartment with time. It remains intriguing why the two developmental phases need two sets of parameters and suggest the existence of a common regulatory mechanism to consider for future models.

### 2.4. Regulations between Thymic Populations

The above-presented models have supposed a certain level of independence between the different cell fates. This is consistent with the essentially linear developmental trajectory of thymocytes from thymus colonization to egress of mature T cells. However, the size of certain thymocyte populations is likely to be subject to constraints, such as availability of survival factors, including cytokines, or cell–cell contacts including interaction with stromal cells or other antigen-presenting cells. The existence of population control or interactions are difficult to validate experimentally. Nevertheless, IL-7 controls overall thymocyte numbers [63,64]. Notably, in the absence of IL-7 or its receptor, the relative proportions of major populations are retained. Consistently, Almeida et al. [65] used murine background models carrying different amounts of DP cells and showed that the number of SP cells were always proportional to the DP compartment size, suggesting that the SP niche is not smaller in the presence of more DPs. Conversely, in conditions of severely limited thymus colonization, such as in CCR7/CCR9 double-deficient mice, population sizes recover to bona fide wild-type levels at the DN3 stage and beyond [66,67]. Recently, it was suggested that at least in a model of cellular competition, the size of thymic populations is controlled through feedback regulation, in which DN2 and early DN3 cells sense DP population size and tune cell-cycle duration in an IL-7-dependent manner accordingly [68]. There is substantial evidence for regulation of mature Treg numbers by IL-2 or IL-15 availability [69]. Competition between T cells for accessing spatially restricted antigens, types of APCs or cytokines could be an additional mechanism balancing the relative amount of each population, and could bring multiple possible fates for thymocytes carrying the exact same TCR, and has not been investigated by mathematical modeling yet. Interestingly, a recent study [70] showed that RAG${}^{GFP-}$ Tregs, resident or recirculating from the periphery, can inhibit the development of newly generated Tregs. We refer to the overview by Klein et al. for details on the complex mechanisms and models for Treg differentiation [71].

Only in some mathematical models, different populations sharing the same ‘niche’ regulate their relative size in a TCR- and antigen-independent manner through a logistic growth control (Figure 2B). Furthermore, the number of cells becoming CD4, CD8, or Tregs are pre-encoded into a differentiation rate instead of a homeostatic control between these populations. The capacity of generational models such as the one established by Thomas-Vaslin to reproduce fast recovery, would argue that a regulation mechanism such as logistic growth is not required *per se* (or at least that logistic growth is not the only possible explanation of the population rebounds). That being said, since this particular generational model inferred a supra-physiologically high proliferation rate for DPs, one would need to assess whether a generational model with different population definitions (between DNs and DP, as discussed above) would manage to reproduce the experimental dataset with physiological proliferation rates. As a rare attempt to model population inhibitions, Kaneko et al. [72] analyzed the kinetics of thymic population dynamics after sub-lethal irradiation that leads to profound but transient atrophy. They compared multiple model structures on how the availability of TEC cells (depleted by irradiation) could regulate other populations (Figure 2B), using iterative fittings [73]. Expectedly, a single ODE could not explain the speed of DP reconstitution and needed a logistic growth mechanism. Furthermore, among the different tested scenarios, the model could best explain the data when DN and cTECs were inhibiting each other’s dynamics. The authors attempted to explain the dynamics of mTECs only from the dynamics of the DP and SP populations and needed to include multiple mechanisms including (i) self-inhibitions of the mTECs and (ii) opposite effect of SP (positive) and DP (negative) on mTEC reconstitution, and/or (iii) impact of DN or cTECs onto DP, CD4 or mTECs. The modeling approach generated 5 possible models explaining well the dynamics of mTECs and the authors selected the most biological consistent with existing literature. This example highlights the complexity of identifying unknown negative regulations between populations from kinetic data. Indeed, the combinatorial number of possible interaction networks is huge, and one could expect that many networks can explain the data equally well. Having many consistent models may help narrow down possible mechanisms and prioritize which ones to measure experimentally. Alternatively, one could use the mathematical model to design a new set of minimal experiments that would be sufficient to discard as many remaining possible explanations (models) as possible, as in [74]. In general, the study of regulation mechanisms might require modeling techniques adapted to their scale, and for instance spatial competition could eventually be best captured using agent-based models instead of population dynamics ODE models.

When comparing the mathematical models for thymus dynamics in (Figure 2B), all models have used a similar and still quite simple structure, despite the large timescale between their design, and the lack of knowledge in developmental stages for the early ones. All models successfully simulated slowly proliferating populations (post-selection DP and SP) with linear ODEs (Figure 2A, left). Regarding models with an additional carrying capacity, one would need to check whether the carrying capacity is actually required on those populations. However, the structure discrepancies arose for highly proliferating (and likely synchronized) populations, such as DN populations and the pre-selection DPs, in which case the model structure has a strong impact and linear ODEs alone are not sufficient to explain the fast rebound kinetics of the thymus after atrophy. The work of Manesso [19] has shown the importance of using generational models for DN1 cells to accurately explain their synchronous development. It is not yet clear whether the later DN populations nor early DP cells show such high level of synchrony, especially since some cells might stay longer at the DN3a/DN3b interface during Tcrb locus rearrangement, and that (although speculative), positive selection could promote heterogeneous proliferation of early DP clones with distinct TCR signaling properties. Proposed size regulations at the DN2 or DN3 stage would suggest including a carrying capacity on these populations. If these cells follow more divisions when the niche is free, a generational model with carrying capacity controlling the number or speed of divisions could be designed, as in [54]. The DN4 and pre-selection DP continuous stages likely harbor the highest complexity in their dynamics, because their distinction is not always clear, the existence of the iSP8 stage, and the possibility of heterogeneous proliferation, which would require better experimental characterization *in vivo*.

In terms of biological rates, the T-cell development parameter values from the models are debatable, and can show high variation between models. We have compiled a list of parameters inferred from experimental data in four major independent modeling studies in Figure 3, normalized to a thymus size of 100 million cells.

As an extreme example discussed above, the identified inflow of DP cells ranges between 0.6 million cells per day (Thomas-Vaslin et al.) up to 30 million cells per day (Sawicka et al.), due to the inclusion of proliferation or not in the DP compartment, showing the impact of model design on parameter interpretation. Furthermore, the size of the thymus is not always mentioned and highly varies between the mouse model and the experimental counting method. Therefore, there is no consensus yet on the best model structure nor most realistic parameters to describe thymus dynamics in general.

## 3. Estimation of *In Vivo* Cell Proliferation in the Thymus

Understanding the strikingly fast dynamics of thymus reconstitution and population size regulation requires the visualization of how fast thymic populations actually proliferate *in vivo* and under perturbations. We have mentioned the work of Manesso et al. [19] and Thomas-Vaslin et al. [53] that estimated the division number from population kinetics. Here we focus on experiments (Figure 4A–D) and mathematical models (Figure 5A–G) aiming at measuring and quantifying the duration of the cell cycle and its phases *in vivo* in the thymus.

### 3.1. Measuring the Number of Divisions by Label Dilution

A first measurement of proliferation involves a dye such as CFSE or CTV that stays in the cell and gets diluted during division. The level of remaining dye in comparison with the original intensity levels thus informs on the number of divisions (Figure 4A). This technique has been rarely used to study thymocyte proliferation *in vivo*, because labeling is performed *in vitro* and thus requires isolation and subsequent transfer into the thymus [75]. Nevertheless, dye dilution approaches have been employed to assess divisions of thymocytes *in vitro*, for instance on a supporting layer of OP9-DL1 cells, or using Reconstituted Thymic Organ Cultures (RTOCs). In particular, Kreslavsky et al. [76] observed that 4 to 5 divisions separated the DN3a/DN3b transition to the entry into the DP compartment *in vitro*, indicating that DN3b, DN4 and iSP8 altogether would contain 4 to 5 divisions. The ETP/DN1 compartment has not directly been checked for number of divisions and Manesso et al. suggested 11 divisions [19]. Finally, Yui et al. [16] observed that ETP, DN2a and DN2b cultured *in vitro* were able to proliferate for 3 to 5 divisions in 3 days depending on the population, but did not check when the cells acquired the next phenotype during these divisions, leaving the possibility of transition to the next population. Meanwhile DN3a and DN2b cells proliferated heterogeneously, whereas ETP and DN2a cells showed a fairly homogeneous proliferation. DN3a cells underwent 2 to 4 divisions before down-regulating CD25 and becoming DN4. Hare et al. [77] showed that the most mature stage of SP4 and SP8 cells can proliferate for multiple divisions in RTOCs under antigen stimuli. Consistently, an *in vivo* study showed that MHC-dependent antigen recognition induced different strengths of proliferation that quantitatively impacts thymic output and might selectively expand some clonal lineages at this late stage [36]. It is not completely clear whether *in vitro* conditions accurately reproduce the *in vivo* signals controlling proliferation, death or emigration (for instance, RTOC cells might not exit and continue proliferating). Finally, Föhse et al. [35] estimated one to two divisions at most from the post-selection DP stage. In general, the number of divisions has been limited to a qualitative constraint for building models rather than being used as a quantitative dataset to estimate parameters.

### 3.2. Nucleoside Analogue Incorporation during S Phase

A second approach is to use EdU or BrdU to label actively replicating cells, as described in (Section 2). We deliberately omit older studies using Thymidine labeling because it was later found to be re-incorporated by cycling cells from dead cells [78]. It has been estimated that BrdU has a half-life of only 12 min in mice and bioavailability of BrdU is lost 60 min after administration [79,80]. Thus, it is well suited for short-term pulse labeling of cells.

#### 3.2.1. Direct EdU or BrdU Staining

Direct EdU or BrdU staining reveals cells that are currently incorporating DNA. It can be used *ex vivo* to label the cells currently in the S phase, or *in vivo* (Figure 4B) to measure the percent of labeled cells (i.e., that were in S phase during the labeling pulse) or the amount of labeled DNA inside these cells, and possibly to track them at later time-points. This technique does not directly indicate proliferation speed nor the frequency of cycling cells, because it does not provide information on the duration of G1, G2 or M phases. For instance, the same BrdU labeling could be generated either by all cells cycling with a long G1 phase, or by only a fraction of cells cycling with a short G1 while the rest would be quiescent. BrdU labeling has widely been used to compare the cycling speed of different populations, but it therefore can be misleading, if the populations have different G1+G2M durations, or if they contain different proportions of quiescent cells. Nevertheless, very low frequencies of labeled cells are an indicator of low proliferation percent or speed (extremely long G1 for instance).

Such methods have revealed that all DN populations are highly proliferating except the DN3a population that is rearranging the Tcrb locus prior to β-selection [81]. Furthermore, among the DPs, mostly pre-selection DP cells, but not post-selection DP cells, proliferate, and only a small fraction of CD4SP and CD8SP cells. Therefore, proliferation would mainly stop before the post-selection DP phase and partially restart in the late stages of single-positive populations.

Altogether, these single-labeling strategies are an indirect way to observe a wave of labeled cells but do not directly capture the details of proliferation (how many divisions, synchronous, and percent of cells dividing). Furthermore, the dilution of signal along with the divisions in the SP stage, as well as the increase in the frequency of labeled cells by division of two half-labeled daughter cells can make the interpretation of results tangled and require mathematical modeling to extract cell-cycle parameters, as done in [82] for population turnover.

#### 3.2.2. One-Point EdU or BrdU Pulse Followed by DNA Staining at Different Time-Points

This approach allows the tracking of the fate or cycle phase of cells that were in the S phase during the pulse at later time-points (Figure 4C). The study by Baron et al. followed the percent of BrdU${}^{+}$ cells after a single pulse labeling *in vivo*, in the full thymus [83]. They observe that DNA amounts linearly increase with time among BrdU${}^{+}$ cells. By linearly estimating the time to reach the highest DNA amounts (4N at G2), they estimated that the S phase would be around 6.5 h. This approach implies to take an average DNA content of all cells in S phase to 3N, because a single BrdU pulse does not allow for the determination of the precise onset of DNA replication in individual cells. By following when BrdU${}^{+}$ cells return to the G1 and to the S phase, they concluded that the G1 duration would be around 10 h while the G2M phase would be of 1.5 h resulting in a full cycle of around 18 h (Figure 4C). Such a fast cycle would be consistent with the fast reconstitution of the thymus after transient atrophy for instance. Since the authors used all thymocytes without gating sub-populations, the results of this study most likely reflect an average behavior among the largest populations of proliferating cells.

Vibert et al. [84] developed a staining protocol, with a first set of 2 pulses of EdU intravenous injections one hour apart, followed by a third EdU pulse 14 h later just before a unique time-point of harvesting the cells, aiming at labeling more cells among slowly proliferating populations *in vivo*. At the time of measurement, the authors additionally stained for DNA content to separate the G0/G1, S and G2 phases together with the EdU labeling. They analyzed in that way three populations: (i) EdU${}^{+}$ cells, i.e., all the cells that were in S phase during at least one pulse. (ii) Cells in G0/G1 that were not in the S phase during the labeling “G0/G1 EdU${}^{-}$“, and (iii) cells in G2/M that were not in the S phase during one of the labeling. They measured aged and young mice of two different backgrounds, for the main populations including separated DN1 to DN4 populations. They built an ODE model for each population with 6 compartments: ‘G0/G1’, ‘S’ and ‘G2M’, each EdU labeled or unlabeled (Figure 5A), and simulated the experimental set-up with instant labeling of the cells in the S phase at the three time-points of the pulses. They inferred the parameters of the model (speed of transfer from each compartment to the next) by fitting the simulations to the three populations at the final time-point of measurement. Obviously, fitting 6 parameters to 3 observed variables at one time-point per compartment was not feasible so the authors took realistic assumptions to reduce the system down to 2 parameters, by limiting death to the G0/G1 stage, by fixing the S phase to 6.5 h from literature [83] (although this value might not apply to all populations), and by neglecting the inflow/outflow of cells from upstream populations during the 16 h of the experiment. This approach raised values of G0/G1 duration from typically 2.5 to 12 days for proliferating populations, while non-proliferating populations such as CD44${}^{low}$CD4SP or CD8SP reached more than 300 days cell cycle (probably an artifact indicating that most of them do not cycle at all). They also observed a lower frequency of labeled cells in 18-month-old mice compared to young mice, consistent with literature [47], and interpreted the results as shorter cell-cycle times in younger mice.

The inferred cell-cycle durations by Vibert’s model [84] are longer in comparison with above mentioned *in vitro* proliferation assays that suggested at least one division per day along DN and early DP stages. Although the model equations were validated by recapitulating the single pulse BrdU kinetics from the study by Baron et al. [83] along a few hours, several factors might need to be accounted for, due to the 14 h period between pulses in [84]. First, some cells could actually have been in two consecutive S phases at first and last labeling (i.e., performing G2, M, G1 and returning into the S phase during the 14 h interval). For the SP populations, bystander non-proliferating cells could help interpreting the low percent of labeling. Finally, there is a possibility that labeled cells from highly proliferating early DP cells could contaminate the late DP compartment that has shrinking dynamics due to high death (i.e., recently arriving cells can occupy a high percent of late DPs at steady state). Finally, the G2/M EdU${}^{-}$ population is supposed to be very minor because most cells at G2M at the measuring time-point were in S phase just before (during the last pulse), which can generate noise in the parameter fitting. Altogether, although labeling more cells, this time-extended experimental setting seemed to generate new layers of complexity in interpreting the labeling results that might require a more complex model design. This example illustrates the complexity of matching a theoretical model with a practical experimental set-up.

#### 3.2.3. Dual Labeling with EdU and BrdU at Different Time-Points

This technique allows differential labeling of the cells entering or leaving the cell cycle, and to follow their cycle phase over time (Figure 4D). For instance, with 1 h difference between EdU and BrdU pulses, this technique has the power to mark synchronized cells entering or leaving S phase at a given interval, and could reveal heterogeneity in the S or G1 phase durations. Thus, Ramos et al. employed such a system to determine alterations in cell-cycle duration of the DN2 and early DN3 compartments suggested to serve as sensors for DP thymocyte numbers [68]. At an excess of DP cells in an experimental model of cellular competition, DN2 cells incorporated less EdU, suggesting that higher amounts of DP thymocytes slowed down the cell cycle of DN2 cells. They then used the EdU / BrdU dual-pulse experiment to build a linear ODE model for cells in S or G phase, labeled or not labeled (Figure 5B), constituting a simplified version of Vibert’s model [84] (Figure 5A). After an EdU pulse, followed by a BrdU pulse at 2 h and harvesting the cells at 4 h, they fit the model with the number of cells in each quadrant.

Using a continuous-time Markov chain model (Figure 5B, right equation), taking into account that some cells can leave and re-enter the S phase during the time of labeling (2 h) while other cells would be extremely slow (which is a consequence of assuming exponential residence time in each compartment), DN2 cells were estimated to have a total cell-cycle duration of 9 h at normal DP thymocyte numbers as compared to 15 h in the presence of excess DP thymocytes [68]. This model was useful in comparing the cycling behavior of cells in two environments for which the EdU/BrdU labeling were already indicative, but additionally providing an estimate of the difference in cell-cycle duration. Notably, an earlier model, assuming that labeled cells cannot return to S phase during the 4 h of staining, inferred very short cell-cycle durations in the range of 3 to 4 h from the same data (Figure 5B, left equation) [85]. This example highlights the impact of model design on the inferred cycle duration values, and underscores that single linear ODEs generate an exponential residence time of cells at each stage, requiring some care in model design or interpretation.

Jolly et al. [86] have proposed an ODE-based model that solves this problem (Figure 5C) by separating each cycle phase into many sequential steps, and applied it on a EdU labeling kinetics scheme in both cell cultures and *in vivo*. This model would also be valid for dual pulse. Due to the complexity of the model, an analytical solution for the dynamics of labeling is not easily available, and a fitting procedure to experimental datasets allows inference of the cell-cycle duration. The equations can conveniently be represented as matrix multiplication and the authors propose an analytical formula linking the frequency of cells expected in each cycle phase with the population parameters assuming steady state growth (also called balanced growth). This approach allows for a reduction of the parameter space or validation of predictions by comparing predicted proportions in each phase to experimental results.

### 3.3. Future Models and Finding the Optimal Experimental Set-Up

The models described above were only partially successful in extracting robust durations of the cell cycle. This might be due to limitations of the datasets that might not contain the appropriate time-points or due to the assumptions of the modeling strategies. One also needs to take into account that the models cannot have more degrees of freedom than the complexity of the datasets to avoid overfitting. Combining all approaches described above a EdU/BrdU dual pulse coupled with DNA labeling at multiple time-points may solve some of these issues [87]. Other modeling approaches could be successful in extracting thymocyte proliferation rates, and in particular how to link the single-cell proliferative behavior to the observed population parameters at the higher scale. Going beyond the ODE modeling approach, or its stochastic continuous-time Markov modeling counterpart (both assuming exponential distributions), age-structured models allow the taking of more realistic time distributions for the cell-cycle duration or its phases, and seem most suitable for cell-cycle simulations.

Recently, Kretschmer et al. [88] studied the cell-cycle duration of memory T-cell precursors and effector cells *in vivo* using the dual EdU/BrdU labeling strategy. Assuming an exponentially growing population, they approximate the relation between the growth rate and the average division time assuming it has no standard deviation. They also derived an approximated mean-field formula of the stochastic model for the number of cells that divided and re-entered the G1 phase (Figure 5D).

In [89], the authors derived analytical formulas for the fate of labeled cells through their progression along the cell cycle. They used an age-structured model where each cycle phase duration follows a delayed exponential distribution (Figure 5E). The authors assumed a ‘balanced exponential growth’ of the population without death, i.e., cells are growing with apparent rate μ (curve proportional to expμt), and kept a constant fraction of cells in each phase over time. The type of chosen time distribution can allow for analytical formulation. Starting from a pool of labeled cells in S phase (just after BrdU), such cells that entered G2M after a time t can be separated as cells of all possible ‘age’ *a* within G2M and therefore the corresponding time δ they took before exiting the S phase since the beginning of (instant) labeling, such that a+δ=t. This is actually a convolution, and using a Laplace transform of the delayed exponential distributions yields an analytical formula for the dynamics of labeled cells either remaining in the initial S phase (Figure 5E, low formula), or progressing to the next phases. Furthermore, the authors provide a formula relating the expansion rate μ to the phase parameters $\alpha_{i}$ and $\beta_{i}$ and the ratio of cells in phase G1, S and G2M: $n_{1},n_{2}$ and $n_{3}$ (Figure 5E, medium formula). They predict that the dynamics of labeled cells from any phase ϕ that progressed to the next phases typically follow two steps: a first period, of duration $\beta_{\phi }$ where labeled cells exit the initial population with a constant speed, followed by a period where the very last labeled cells exit, revealing the exponential decay part of the S-phase duration distribution. The authors show that the initial derivative of the curve requires two early experimental points and is enough to set the expansion rate and some alpha parameters, while a third experimental data point is needed after $t=\beta_{\phi }$ to identify the average duration of the exponential decay. This approach therefore seems suitable to interpret *in vivo* thymocyte EdU/BrdU labeling, with the limitation that the third optimal experimental time-point is difficult to estimate because it needs a pre-existing guess on after how long the cells in S phase start to leave (time $\beta_{\phi }$), and the exponential decay might be very short. Since the model has been designed for cells growing in culture, it is yet to be determined whether the hypotheses of no death and balanced growth would still be valid *in vivo* where cells can exit a compartment, potentially after a regulated number of divisions.

Zilman et al. [90] proposed a more general age-structured model including a distribution of inter-mitotic time (cell-cycle completion) and death, derived from the von Voerster equation [91], which relates the number of cells and their age within a population as a partial differential equation. More precisely, the distribution of the age of cells within each generation is stored, and evolved at each time-point. The fate of the cells at the next time-point is a convolution of cells at each age and the distribution of time inside this generation (inter-mitotic time) or death. Again, a Laplace transform becomes convenient because it transforms the convolutions into multiplications (Figure 5F). The authors derive analytical formula for the dynamics of a pool of labeled cells and reproduced quite well experimental datasets using labeled dye dilution *in vitro*. Although models for T-cell proliferation have typically used lognormal distributions [92], the authors show here that the gamma distribution could successfully reproduce the experimental data and is therefore also a suitable distribution to study cell-cycle progression, especially as the lognormal Laplace transform is too complex [93]. The authors also adapt their formula to branching imbalanced divisions allowing the introduction of asymmetric divisions.

Altogether, it is likely that a combination of Weber et al. approach [89] with the generational model of Zilman et al. [90] including death could allow the derivation of analytical formulas for BrdU or EdU labeling that fit with synchronized proliferation with a fixed number of divisions in the thymus and be used for *in vivo* experimental datasets.

A last and most general strategy is the explicit simulation of the stochastic equations using an agent-based model with thousands of cells with an associated distribution of time for each event (Figure 5G), as done for 2D tumor tissue cell cycle in [94]. Each cycle phase can follow a lognormal distribution as in the cyton model [92,95], and death can be drawn as an exponential distribution, or could be restricted to the G1 phase for instance. It becomes easy to simulate the exact experimental setting.

Future technical development might guide the design of new types of models, such as for the interpretation of Ki67 expression [96,97] and its degradation at specific cycle phases. The measurement of TREC recombination circles dilution from TCR recombination is an indirect read-out for proliferation and population dynamics that has been leveraged using mathematical modeling [98] and is suitable for analyzing human samples as well as the use of labeled deuterium in drinking water [99]. Similar to the dilution of labels introduced *in vitro*, dilution of genetic markers may serve as measures for proliferation *in vivo*. Thus, such experiments circumvent potential biases introduced by varying culture media or unfavorable treatment during isolation. RAG recombinase is stage-specifically expressed in thymocytes undergoing somatic recombination of TCR genes and rapidly shut-off thereafter. Thus, using RAG1${}^{GFP}$ reporter knock-in or transgenic strains, dilution of GFP serves as surrogate for proliferation after termination of TCR gene rearrangement [60,100,101]. To overcome the need for normalization to correct for degradation of GFP in this experimental system, the half-life of GFP has been prolonged to weeks or even months by fusing it to histone 2B [102,103]. Such fusions have been used to generate Tcrd-H2B-GFP mice to label γδT cells [104]. During recombination of the Tcra locus, Tcrd and thus H2B-GFP coding sequences are excised and protein expression ceases, making H2B-GFP levels virtually exclusively dependent on dilution through proliferation. This system has been used to analyze dynamics of various αβT-cell populations [35]. Dilution of genetic labels is based on minimal perturbation of cell state. Therefore, experiments relying on such approaches are likely to provide excellent data for future mathematical modeling. Finally, newly developed *in vivo* reporters for cell cycle might allow more precise longitudinal evaluation of cell cycle over time [105].

As a conclusion, the mathematical models for labeling-based cell-cycle dynamics inference mainly differ in their assumed time distributions of the cell-cycle phases. ODE-based models suffer from implicitly assumed exponential time distributions, allowing very few cells to artificially cycle extremely fast. Although ODE models could raise a good fit to experimental data with very scarce time-points, it is not sure whether they would manage to reproduce time-resolved datasets with many time-points. This can be solved either by duplicating each phase into sub-phases as in Jolly et al. [86] and performing parameter optimization, or in the general case by considering age-structured models. Age-structured models offer analytical formula under balanced growth, or can be simulated as PDEs or agent-based models, with higher flexibility to account for experimental biases (efficiency of labeling or duration of the label pulse; possibility of inflow of labeled cells from progenitors if the experiment timeframe is long).

A recommendation for future experimental-modeling hybrid approaches would be to perform test experiments with many time-points to assess whether the model structure is suitable (possibly *in vitro*, as in Jolly et al. [86]), and which time-points are most informative (as suggested in Weber et al. [89]) before analyzing *in vivo* data-points with reduced time resolution. The discrepancy in gating strategies, in particular the boundaries between DN4, pre-selection DPs and post-selection DP, would be beneficial to be provided, because it forbids to compare the results between studies. Most studies neglected the iSP8 population, whose high proliferation might artificially contaminate the inferred proliferation rate of SP8 cells, and would be beneficial to separate as an additional population. Finally, it appeared that the dynamics of labeling is not always enough to infer the full cycle duration (as in [86]), possibly because with few time-points they can only inform on the ratio between each phase, rather than the absolute cycle duration. Such studies additionally used the amounts of cells at steady state in each cell phase to infer the cycle duration.

So far, all models assumed homogeneous population behavior and did not account for a higher heterogeneity of the cell-cycle duration (for instance the possibility of bimodal time distributions where some cells need to wait for a proper Tcra or Tcrb recombination to proceed), as well as the impact of population synchrony (if assuming a fixed number of divisions in a compartment impacts the interpretation of the labeling). They are likely next topics to be investigated when experimental techniques will allow for a better description of cell heterogeneity.

## 4. Estimation of *In Vivo* Cell Death in the Thymus

A substantial number of thymocytes fails quality control imposed by positive and negative selection. Estimating the rates of thymic selection is critical for the calibration of mathematical models of T-cell developmental dynamics. However, cell death is particularly hard to visualize *in vivo* and macrophages can remove thymocytes extremely fast and even seem to contribute to inducing cell death [106]. Experimental approaches to determine the extent of thymic selection, sometimes combined with mathematical modeling, have been reviewed in [6]. We provide a brief overview here, illustrating some key experimental constraints. Of note, depending on the study, the ‘efficiency of selection’ can be estimated either as flow of cells dying per day at a certain stage (rate), or as the number of cells that will die or survive through selection from a defined pool of cells (percent). The latter definition depends on the residency time of cells at different stages, which is also hard to measure for heterogeneous populations. Several early studies estimated rates of selection by either directly inducing negative selection [107] or removing selecting ligands (i.e., MHC) from a variety of thymic APCs to induce failure of positive or negative [108,109,110,111]. Together, these studies yielded a broad range of frequencies of death by neglect or clonal deletion. However, interpreting these data is difficult, as removal of MHC removes both positively and negatively selecting signals. Moreover, clonal deletion may occur throughout the differentiation process, ranging from DP thymocytes that have just completed Tcra rearrangement to SP thymocytes, allowing for multiple interactions of a thymocyte auditing for selection and different types of thymic APC. Another approach was based on continuous BrdU labeling using transgenic T cells with CD4 or CD8- biased TCRs that were known to survive positive selection [112]. The aim was to monitor the maximum number of cells that could survive through positive selection *in vivo* by filling the thymus by survivable TCRs and compare this number to that of surviving cells in the WT setting. This study suggested that at least 40% CD8 TCRs and 90% CD4 TCRs are removed through both positive and negative selection combined.

More recent studies to estimate rates of selection rely on one of two broad approaches: (1) use of TCR signaling reporters based on the hypothesis that negatively selected cells have received TCR signals of higher affinity and (2) analysis of the outcome of TCR/pMHC ligand interactions.

Two studies have revisited death estimations using signaling reporters. Stritesky et al. used a Nur77${}^{GFP}$ reporter to quantify levels of TCR signaling in thymocytes [28], comparing WT or Bim-deficient mice, in which negatively selected thymocytes fail to undergo apoptosis. The authors distinguished three populations based on GFP reporter expression: GFP low cells that die by neglect (positive selection), GFP intermediate cells that have received a positively selecting TCR signal, but may still audit for negative selection, and finally GFP high cells that are deleted in WT mice but persist in Bim${}^{-/-}$ cells. Following the observation that Bim${}^{-/-}$ cells spend longer in the SP4/SP8 compartment than WT cells on average, they estimated that at the scale of a 200–250 million cells per thymus, 3 million cells survive both positive and selection per day, while 16.7 million cells would die by negative selection. A minor caveat for determining exact rates of selection stems from the observation that Bim${}^{-/-}$ thymocytes have an increased residence time when compared to WT cells in the SP compartment, because they do not die and are kept longer in the thymus. However, Bim${}^{-/-}$ cells comprise both GFP intermediate positively selected cells, which should exit normally as WT cells, as well as GFP high cells, which are indeed staying longer. As raised by Yates [6], dying cells and surviving cells have a different residence time (even if following the same mechanism). This means that extra Bim${}^{-/-}$ cells that “should have died” stayed actually longer than the average residence time of all Bim${}^{-/-}$ cells, and negative selection could therefore be slightly lower than estimated.

Daley et al. [113] used a similar approach based on accumulation of cells poised for clonal deletion in Bim${}^{-/-}$ mice and analyzed the outcome of TCR/pMHC ligand interactions in a dual transgenic TCR/cognate antigen model. Expression of self-antigen deleted 60% of the CD4 SP cells compared to mice without expression, while in Bim${}^{-/-}$ cells, those cells survived. The authors identified Helios as a surrogate marker for cells undergoing negative selection. Using this marker in combination with markers of progressive thymocyte maturation, they proposed a multi-step model of clonal deletion, concluding that negative selection deletes 55% of the positively selected thymocytes already in early SP cells. More individual TCR interactions and their outcomes were probed in a recent paper by MacDonald and colleagues [114]. Using parallel cloning of multiple TCRs and analysis in retrogenic mice, this study showed that of all analyzed TCRs, 85% failed to be positively selected and of the remaining 15%, half were subjected to negative selection, essentially in line with other estimates. Although the design of this study does not allow a conclusion regarding the temporal sequence of selection, it provides valuable information on the nature of positively and negatively selected TCRs. Notably, the latter displayed a higher degree of cross-reactivity.

Finally, some population models described above, such as those developed by Sinclair et al. [43] or Thomas-Vaslin et al. [53] inferred death rates from their experimental datasets, but from populations lacking proliferation. This means the inferred rates are actually including the effect of proliferation, and could be re-estimated based on proliferation studies. Sinclair estimated that 75% of cells fail positive selection and only 2 to 5 percent of cells become CD8 and CD4 at the end, respectively. Including proliferation at SP stage would actually mean that more cells died by negative selection, probably not that far away. In Thomas-Vaslin’s model, where cells can die only at the DP stage, 97.5% of the pre-selection cells die at that stage.

Taken together all studies converge on a very high frequency of death through selection, between 90 to 97.5%, which could be even higher when including proliferation. However, it remains a challenge to fully disentangle the contribution of death by neglect vs. clonal deletion as well as the type of APC, onto this death load. In conclusion, a thorough comparison of experimental datasets ranging from signaling reporters, dynamical datasets (like recovery after atrophy), and EdU/BrdU labeling into a single mathematical analysis could narrow down the selection rates with better understanding on the experimental perturbation biases, yet is very complicated.

## 5. Multi-Scale Considerations on Thymic Dynamics

Selection processes in the thymus constitute quality control mechanisms downstream of the bona fide random somatic recombination of TCR genes into a functional but not self-reactive repertoire. Thymic selection emerges from events at the molecular and cellular level (Figure 6A). Understanding how the dynamics of T-cell development arise from these lower scales requires multi-scale modeling.

At the molecular (and genetic) scale, virtually each thymocyte that completed the pre-selection DP stage, carries a different somatically recombined TCR, composed of one TCRα chain and one TCRβ chain at its surface. Lack of allelic exclusion of the Tcra locus allows for the generation of T cells with two distinct TCRs. APCs display a sub-sampling of possible self-peptides on their surface MHC complexes. Binding between TCR complexes and pMHC complexes triggers TCR signaling on the thymocyte. The landscape of self-antigens presented in the thymus is particularly complex as it depends on the type of APC, their capacity to express many proteins from the genome, distinct mechanisms of antigen processing, and the structure of the 6 MHC proteins encoded by highly polymorphic genes.

At the cellular level, thymocytes move and sequentially interact with APCs. The multiple pMHC complexes and TCRs of the APC and thymocyte, respectively, located in the membrane cell–cell contact, have the possibility to interact. The affinity (existence of high-affinity binding) as well as the avidity (amount of binding TCR-pMHC couples) is translated into TCR signaling that is integrated between cellular contacts.

The ultimate outcome of thymic selection is defined by a TCR repertoire on peripheral T cells, which can recognize foreign peptides (antigens) in the context of self-MHC, and the bona fide absence of cells whose TCR form high-affinity interactions with self-peptide loaded MHC. It is not fully clear how positive and negative selections are decided, depending on the TCR affinity, TCR cross-reactivity to different self-peptides, and the avidity of sequential cellular interactions, through TCR signaling [34,114,115]. Finally, the boundary between negative selection and Treg-cell differentiation is unclear as both Tregs and Tconvs surviving thymic selection share some identical TCRs (see the overview of Klein et al. [71] for a review of Treg differentiation models). Several multi-scale mathematical models predicted the properties of the produced TCR repertoire due to positive and negative selection, based on a static set of TCRs and MHCs. These models, comprehensively reviewed in [6], have been helpful in particular to understand trade-offs between TCR cross-reactivity, pathogen recognition and auto-immunity; the induction of MHC recognition, restriction or Treg differentiation from positive and negative affinity selection thresholds; or how thymic selection generates holes in the repertoire for pathogen coverage. Very few models however have investigated how thymus dynamics arise from the lower scale of single-cell motility and fate decision, and how it affects the higher scale of repertoire generation and TCR clonality.

### 5.1. Linking the History of TCR Signaling to Cell Fate

Several studies have proposed to link the dynamics of TCR signaling to thymic selection processes, which would constitute suitable bases to simulate thymus dynamics in the future.

First, Grosmann et al. [116] proposed a theory on how the dynamics of TCR signaling could look like, and how it could be translated into positive or selection decisions (Figure 6B). Based on the observation that TCR expression and signaling gradually increase over the DP stage the authors proposed that T cells maintain two tunable activation thresholds: a lowest signaling level threshold to survive positive selection, and a higher threshold to delineate deletion by negative selection, and that both thresholds would adapt to each other or to the current signal level. They defined a variability-maintenance threshold that grows together with the expression of TCRs at the surface, and an activation threshold, linked to the former threshold, relatively higher than the maintenance threshold, for negative selection. This theory can explain that DP and SP cells have different activation thresholds, and allows T cells to self-tune their signaling thresholds to their environment (antigen expression pattern), possibly giving a robust response independently of a constant inflammatory context or perturbation. A large body of evidence has detected quantitative and qualitative differences in the TCR signaling leading to positive or negative selection [29,30,31,32,33]. Bhakta et al. [117] and a series of papers from Ellen Robey’s lab [118,119,120,121] could directly visualize the temporal signals received by thymocytes during positive and negative selection, thanks to *ex vivo* calcium imaging of thymic slices (Figure 6C). Calcium signaling was observed, composed of peaks of typically 5 min interspersed by 25 min of resting, matching patterns of stop and migration, while encounter of cognate ligand led to pronounced arrest and elevated levels of sustained signaling, eventually resulting in cell death. The observation that positive selection happened in *ex vivo* 3D slices but not in *in vitro* cultures [121], support the hypothesis that transient regularly interspersed signals are required for proper signal-to-fate decisions. It is plausible that a cell needs to regularly detach from one APC to the next to avoid strong signaling. This is a rare study linking calcium signaling to changes in motility suggesting that the patterns of T-cell search are also impacted by TCR signaling and could benefit from a modeling on their own.

An agent-based model has been developed by Khailaie et al. [122] (Figure 6D) to link cell–cell interactions with single-cell TCR signal integration, using string models for TCR-pMHC affinity with short range positional correlations. In this model, a list of T cells with random TCRs sequentially interact with APCs carrying a random sampling of pre-defined self-peptides at each time-point, all presented on the same MHC molecule, and a TCR signal is integrated over time at each interaction with a decay rate. This leads to peaks due to encounter with higher affinity peptides with a constant contribution of the MHC at each interaction. Similar to the studies of Grossman et al. [116] and Kurd et al. [121], the authors proposed to use a threshold on the basal signaling level of a thymocyte as a decision to survive positive selection, while a threshold on the highest peak would define autoreactivity of a TCR. Khailaie also noted a trade-off for selected cells between sustained and peak signaling (cells with higher basal peak would die when they encounter a medium affinity peptide while low basal peak would allow the binding of peptides with higher affinity and survive), and suggested that Treg cells are more affine to MHC, which would endorse them with higher cross-reactivity.

Recognition of self-peptides plays a substantial role in positive selection [123,124,125,126], but their relative abundance is heterogeneous [127]. It is tempting to propose that frequent or groups of structurally similar antigens could generate a signal supporting positive selection of TCRs recognizing them with medium affinity, while rare antigens would not have this capacity. It would be interesting to check whether this happens in Khailaie’s model using a mixture of frequent (possibly similar) antigens and more rare antigens, and correlating cell fate with affinity to these frequent antigens instead of MHC affinity.

Although the link between basal and peak signaling and positive and negative selection, respectively, is still speculative, different signaling pathways have been associated with both selections and Treg-cell development (for instance ERK and Ras signaling [32,33]). It is tempting to hypothesize that different pathways could behave as band-pass filters, with some ‘fast’ pathways getting activated by rare but strong encounters (peak signal) and other pathways slowly activated after many repeated interactions [29]. For instance, an ODE-based model for TCR signaling has been predicted to be able to discriminate self and foreign peptides [128] based on interaction frequency. Light-controlled activation of TCR signaling as shown that TCR signaling can discriminate signals based on their frequency [129]. More generally, in the context of frequential inputs, a Fourier transform of TCR signaling models could help predicting the signal properties required for different fate decisions.

Together, these studies support the notion that positive and negative selection could be mechanistically defined by different types of signaling, in which a rapid, high peak promotes negative selection while basal signaling defines positive selection. Notably, the experimentally observed signals were in almost perfect agreement to those predicted by [116]. Furthermore, these models are suitable to simulate developmental dynamics at the DP and SP stage from single-cell encounters, although at present these models cannot yet incorporate a possible interdependence between TCR signaling and a cell’s decision to proliferate at the early DP or late SP stages. In contrast to single-cell models, dynamical models presented in Section 1 encoded differentiation (to CD4SP, CD8SP or Treg cells) as a constant rate, thus masking potential regulatory mechanisms that agent-based models would intrinsically include. For instance, Kurd et al. [121] suggested that mismatched CD4SP or CD8SP with a TCR that recognizes the ‘wrong’ MHC would not get sustained signaling in the SP stage, and die by neglect, showing that signal integration is likely also important for CD4 vs. CD8 differentiation dynamics.

### 5.2. 3D Models of Thymic Development, APC Types and Antigen Spatial Compartmentalization

T-cell development is coupled with regulated migration patterns. ETPs enter at the cortico-medullary junction from blood vessels, and both DN and DP development happen inside the cortex, where cTEC and other APCs support positive and negative selection of DPs. The maturation of DPs into SP is associated with changes in chemokine receptors and migration towards the medulla that occur typically 12 to 24 h after the onset of positive selection [121]. CD4 and CD8 T cells down-regulate the opposing co-receptor with different timing, later for CD8 SP cells. In the medulla, AIRE-expressing mTECs show a larger panel of antigens referred to as tissue restricted antigens (TRAs), while other APCs (dendritic cells (DCs), B cells, stromal cells [130]) can also present self-antigens, or antigens captured in the periphery by migratory DCs [71,131,132]. For instance, DCs seem to be more efficient at mediating negative selection in the cortex while cTECs are also presenting self-peptides. Rare DCs are located close to capillaries and surrounded by CCL21. Interestingly, DCs and mTECs as well as vascularization are much denser at the cortico-medullary interface, suggesting its crossing has the highest strength of selection (or avidity) [71]. The molecular cues guiding the transit of thymocytes from cortex to medulla are poorly understood. Although it is well established that chemokine receptors CCR7 and CCR4 play dominant roles in this process, it has recently been suggested that these receptors indirectly promote spatial organization of thymocytes by organizing the localization of thymic APCs, in particular DCs, and mediating their interaction with thymocytes [133,134]. Consequently, loss of either chemokine receptor results in defects in central tolerance [134,135]. Notably, the presence of a thymic medulla is critical for development Treg, but not Tconv, cells [58], suggesting that Treg cells may more stringently depend on TRAs or medulla-specific cofactors. This added complexity may have implications for future modeling approaches. The egress of T cells is mediated by the S1P1 receptor, which is up-regulated only at the latest stage of SP maturation. Treg cells are believed to stay longer in the thymus, and different types of APCs harbor different Treg inducing capacity [136]. It has been suggested that certain antigens could be expressed in spatial niches, added to the fact that TRAs are preferentially expressed in the cortico-medullary junction [71]. These arguments support the notion that the spatial organization of the thymus is critical for its function and would call for spatial mathematical models.

So far, only a few models have attempted to simulate the thymus in 2D or 3D. Elfroni et al. [137] developed a framework able to simulate motility, chemokine sensitivity, proliferation and death on a 2D lattice system, similar to later developed platforms such as Morpheus [138], and including the cortex and medulla. The model was used to recapitulate the migration of cells following chemokine gradients [139], in the context of WT versus CCR9-deficient mice. They also measured an effect of space competition into apoptosis and the subsequent amounts of generated CD4 and CD8 T cells. Furthermore, the cell–cell contact dissociation rates impacted the CD4 to CD8 decision. Souza-e-Silva et al. [140] used a simpler mathematical formalism and simulated chemokine levels and cell decisions as a cellular automata, i.e., a 2D grid, where each position can only have pre-defined states that are updated according to the neighboring states. Interestingly, the authors could reproduce realistic movement between cortex and medulla, and proper dynamics of development and residence times from a few cells to a full thymus at equilibrium. They could modulate the T-cell dynamics from changing the properties of the epithelial network. The fact that correct dynamics can emerge from simple 2D models makes it tempting to believe that it will soon be possible to incorporate multi-scale models in a 3D setting, incorporating data on migration from thymic slices, proliferation, population dynamics and signal integration. The findings that thymocyte migration and signals are correlated [121] would suggest using such models to calculate a signal integration and fate decision from the interactions such as [141,142], feeding back to a modulated searching behavior of the cells.

### 5.3. Thymus Dynamical Models Can Help the Analysis of TCR Repertoires

High-throughput sequencing has provided in-depth information on TCR diversity generated in the thymus [143]. Dynamical models of T-cell development are likely to help understand the formation of pre-selection repertoires and their shaping through selection.

For instance, a mathematical model has been developed to simulate V(D)J recombination of the Tcra or Tcrb gene [144]. This tool takes a repertoire and proposes the most likely V(D)J recombination event for each sequence by inferring probabilities of using each V, D or J segment (called recombination parameters), including deletion and insertion events. Then, from inferred recombination parameters, it becomes possible to generate new TCR sequences following the same recombination model. Although this model has been used to analyze peripheral TCR repertoires, it actually simulates Tcrb recombination before the DN3b stage or Tcra recombination during the DP stage, which are impacted by both thymic selection and population dynamics. An example in Figure 7 demonstrates that proliferation can strongly alter the relative frequency of clones in the periphery. We speculate that using knowledge or models on thymus population dynamics, new recombination models could be designed, which include variation in clonal expansion for the analysis or generation of TCR repertoires. Furthermore, such models comparing pre-selection and post-selection repertoires could identify which TCR sequences are preferentially deleted or expanded, as attempted in [145]. Overall, the existence of public TCR clones (sequences shared between individuals) has been a challenging phenomenon to explain from recombination models that predict lower probabilities for each TCR sequence, and might well come from selective clonal expansion in the thymus.

Moreover, the patterns of thymocyte clonal expansion are poorly described and it is not clear whether T cells are selected independent from each other, and to which extent there is competition between antigen-specific clones, or how the competition for cytokine signals determines terminal differentiation. Newly developed *in vivo* barcoding approaches coupled with statistical analyses are suitable to follow the clonality of progenitors along thymic selection [146]. Such datasets will likely support the development of lineage tree algorithms possibly combined with population dynamics and branching fate decisions in the thymus. In turn, this could provide valuable information on the relative clonal expansion in DP and SP stages to simulate proper population dynamics.

### 5.4. Future Types of Multi-Scale Models

New experimental datasets such as single-cell RNA sequencing showing both population delineation by transcriptomics and receptor sequencing [147] will surely unveil new properties of thymic selection, reveal new hidden sub-populations and link cell fate to their TCR sequence.

Recently, a multi-scale dynamical model has been developed covering the earliest steps of intrathymic T-cell development until completion of lineage commitment (i.e., the DN2b stage) [37]. This agent-based model comprises gene regulatory networks, epigenetics and population dynamics based on single-cell gene expression data for key transcription factors as well as *in vitro* differentiation and proliferation dynamics of populations and individual clones. Experimental data had revealed that expression of the T-lineage commitment Bcl11b is subject to complex regulatory mechanisms involving an interplay of cis-acting and trans-acting elements in combination with a degree of stochasticity [148]. Furthermore, simple gene regulatory networks were not able to fully explain observed and modeled population dynamics of immature thymocytes, which we extensively discussed in Section 2. This type of multi-scale model allows for a smooth transition between differentiation stages, going beyond the ’yes-or-no’ gating strategies shown in Figure 1A, and could reveal unexpected cell conversion or differentiation pathways in the future. Together, this study highlights the requirement of extensive and complementary datasets to build a mathematical model with sufficient explanatory power as well as the requirement for multi-scale approaches to adequately represent the increasing level of detail of our molecular, cellular and organismal knowledge of developmental processes.

## 6. Outlook

We aimed to provide a broad overview of scales of intrathymic T-cell development that have been simulated by mathematical modeling, where the same mechanisms are treated from different angles and data types. Modeling has been necessary to infer hidden experimental information at the cellular (proliferation speed, death) or population scales (developmental dynamics), or to understand fundamental properties of this complex selection system. The next generation of models will most likely include multi-scale datasets, such as proliferation speed, population dynamics, as well as signaling, and will likely be single-cell-based.

Modeling intrathymic developmental dynamics is limited by in part scarce experimental data on numbers, developmental trajectories, dynamics or even function of non-thymocyte populations. The role of dendritic cells and TECs in selection is well characterized, including part of the molecular basis of cell–cell interactions with thymocytes. Accordingly, especially TECs and their interactions with developing T cells have been started to be incorporated into models of thymus regeneration [72]. However, developmental trajectories of dendritic cells and TECs remain a matter of debate [149,150]. The function, including their interaction with thymocytes, of other cell types in the thymus, including B cells, eosinophils, innate lymphoid cells, recirculating mature T cells or non-epithelial stromal cells, remains to be clarified [151,152,153,154]. Thus, more experimental data is required before models can faithfully incorporate such cell types.

Another limitation due to the virtual absence of data constitutes the connection between cell-cycle control and circadian clocks. Both systems share common regulators and interdependence of both has at least been suspected for some biological systems [155,156]. Given the vast body of extant experimental data, T-cell development may in fact serve as an excellent experimental model to address this interplay both experimentally and mathematically.

Although we focused on the cellular and population level mechanisms that have been modeled in the context of thymus dynamics, a next challenge will be to understand and link the molecular determinants of cell decision into modeling T-cell development. The study by Olariu et al. [37] is a good pioneering example of how a ‘full understanding’ of single-cell differentiation can be reached using gene regulatory networks. Such molecular-based models could help understanding the role of transcriptional regulation and non-coding RNAs on population scale T-cell fate decisions. The complexity of TCR signaling dynamics, which has extensively been modeled [157,158,159], could benefit positive or negative selection models, including stochastic noise on TCRs expression [160].

T-cell developmental dynamics as a model system highlights the diversity of modeling methods used at the same scale, and the complexity to measure and bridge cellular events to population dynamics. ODE models are not always best suited, due to their assumption that cells stay with an exponentially distributed time before leaving, and might require special care in their design or interpretation. Generational models can explain fast thymus recovery after atrophy, as well as logistic growth mechanisms, suggesting that controlled number of divisions is a potential homeostatic mechanism for a fast thymic reconstitution. Dynamical models were powerful at hypothesis testing, by providing the most suited mechanistic scenario to explain the datasets, but were poorly able to identify biological parameters from experimental datasets. Underlying reasons were either parameter uncertainties (only a few studies actually showed identifiability of their parameters) or model uncertainties (that another model structure would cause the model to infer different parameter values). Rigorous testing of different possible model structures requires a dissuasive amount of work. Consequently, no consensus on the proliferation, death (although quite close) or differentiation rates during T-cell development has been reached, despite the large extent of datasets. Given the diversity of datasets and complex experimental setups, these datasets remain difficult to combine. A more general question would be: what is missing from these datasets to finally infer these biological rates? Or which perturbations would be needed to actually identify the strength of competition or regulation between populations? The panel of models reviewed here may provide cues to design next-generation multi-scale models.

## Figures and Tables

**Figure 1 entropy-23-00437-f001:**
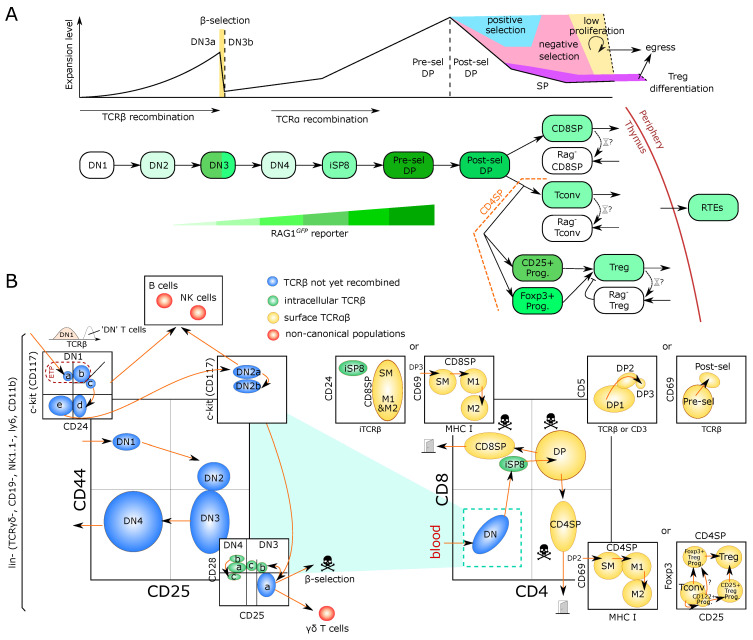
Major developmental steps in the thymus as a basis for population models of T-cell development. (**A**) Main stages annotated with their degree of expansion (y axis) and RAG1${}^{\mathrm{GFP}}$ reporter expression (green levels). In such an experimental model, RAG1 gene regulatory elements drive expression of a reporter gene, such as GFP, the concentration of which depends on cell division and reporter protein half-life. Thus, reporter levels can be used as a timer and distinguish newly generated versus recirculating or long-term populations. The main bottlenecks in transition between thymocyte populations are β-selection, selecting for cells with functionally recombined TCRβ, and positive and negative selection that select for cells with functional MHC reactive, but not self-reactive fully expressed TCRαβ. The hourglass denotes that loss of RAG expression could also come from cells being resident in the thymus for a long period of time (instead of recirculating), which is an open question. (**B**) Gating strategies of functional sub-populations. The first lineage gating ‘lin-’ on the left discards B, NK and myeloid cells. (left) Detail of developmental stages inside the DN population, and a choice of markers to distinguish them. Progenitors inflowing from the blood are called Early T-lineage Progenitors (ETP) and refer to DN1a and DN1b. DN1 and early DN2a cells can also differentiate into B or NK cells while only late DN2bs are fully committed to the T-cell lineage [16]. When the DN4 population is only gated on CD4${}^{-}$CD8${}^{-}$CD28${}^{-}$CD44${}^{-}$, it also contains more differentiated populations containing TCRβ [17]. (right) Main developmental stages from the DN stage to fully mature CD4 and CD8 T cells and their export (open door symbols). Different gating strategies are shown for isolating DP and SP sub-populations. The death skulls refer to stages with high death. The term Tconv refers to conventional CD4${}^{+}$ SP, while CD8${}^{+}$ SP cells can also contain conventional and unconventional cells that are not described here. The relative size of each compartment is detailed in [17].

**Figure 2 entropy-23-00437-f002:**
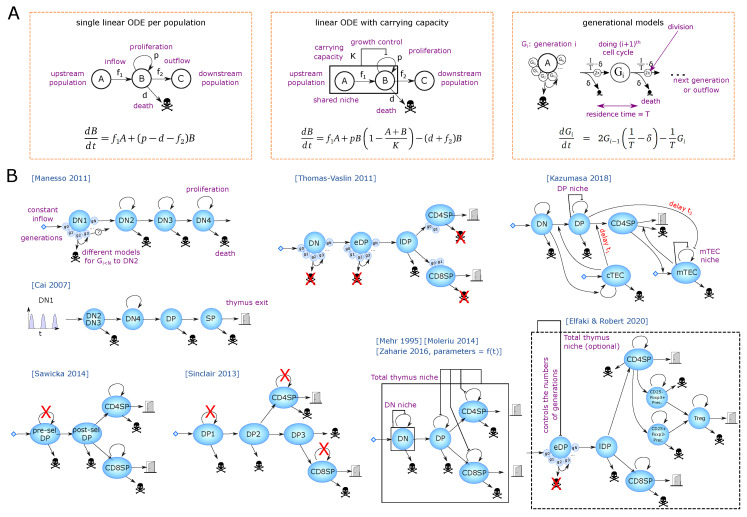
Population dynamics mathematical models of the thymus. (**A**) Types of equations used when simulating thymic population dynamics, accounting for the dynamics of a population B fueled with progenitors coming from a population A, and further differentiating into a population C. (left) simple linear ODE with proliferation (round arrow), death (death skull) and differentiation (flat arrows). (middle) linear ODE with an additional regulated logistic growth according to a maximum carrying capacity *K*, and whose niche is shared with another population A (large box). The logistic growth control in thymus models has been implemented by inhibiting proliferation rather than enhancing death. (right) linear ODE-based generational models that simulate the cell numbers at each division within the population A. $G_{i}$ denotes the number of cells inside the generation *i*, i.e., that performed *i* divisions already. The rate of cells leaving a generation is 1/T where *T* is the half-life of a generation, and the rate of cells entering the next generation is 2(1/T−δ) where δ is the death rate. This type of model assumes a generation-structured population behavior, i.e., that all cells perform a fixed number of divisions before exiting the compartment A, which can generate different dynamics than the linear ODE model on the left. It is also possible to add an outflow rate at each generation to change this behavior (not shown in the formula, see model variants in [19]). (**B**) Published mathematical models, annotated with the equation design explained in A. Death skulls refers to a linear death rate, round arrows refer to proliferation, the large boxes represent a carrying capacity, while smaller sub-populations in circle denote a generational model. The red crosses denote neglected mechanisms in the models, and the open door refers to a linear outflow rate.

**Figure 3 entropy-23-00437-f003:**
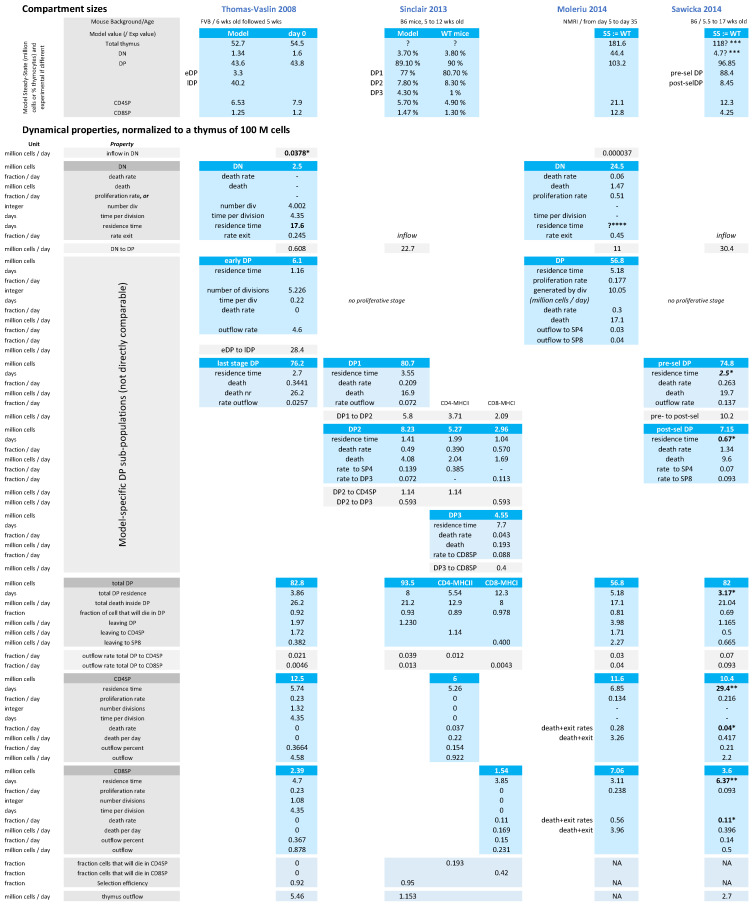
Parameters from four main studies [43,45,53,55]. (**Top**) The size of each considered population is shown, at steady state in the models. Sometimes the model stabilizes at a different value than the experimental dataset, in which case the experimental value is given for comparison. All cell numbers are in million cells. (**Bottom**) Detail of model parameters and cell numbers. All absolute values (cell numbers or flow between compartments) are rescaled to a total thymus size of 100 million cells, to be more easily compared. Technical details: Parameters from Sinclair et al. [43] are average values digitized from its Figure 3, under the “4+8” model, and the details of DP2/DP3 sub-populations are calculated from percentages shown in its Figure 7. The “parameter set 2” is shown for the study by Moleriu et al. [55]. *: this value was taken as a hypothesis and was not inferred from experimental data. **: we calculate residence time as 1/(output + death − proliferation), which is the half-life of the population dynamics. The authors instead calibrated the half-life of one cell (excluding its potential daughters), as 1/(output + death), to match experimental data, which ended up as a very long population residence time here. ***: This study did not show the number or percent of DN cells. We assumed a DN population size of 4% of the thymus to estimate the total thymus size and rescale the cell numbers to 100 million cells. ****: the calculated residence time diverged, probably because of digit precision on the parameters.

**Figure 4 entropy-23-00437-f004:**
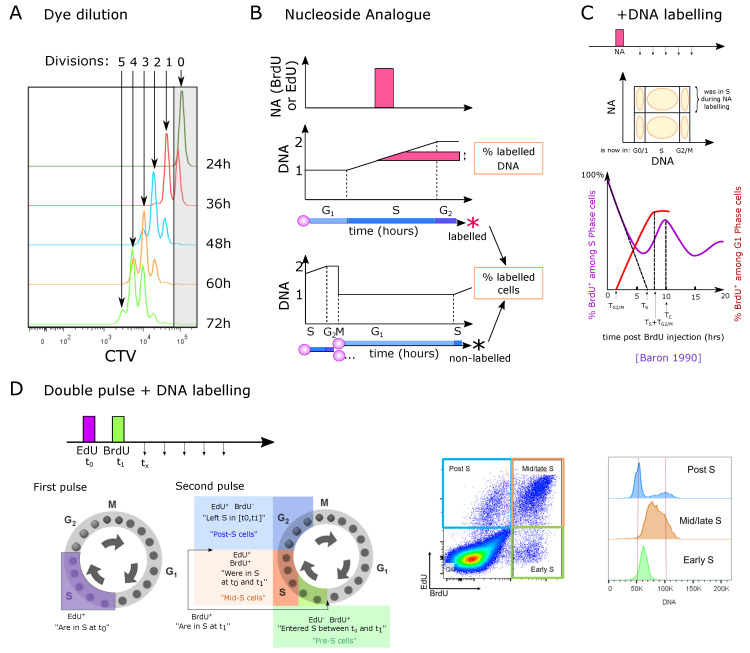
Experimental methods to measure proliferation in the thymus. (**A**) Following the number of divisions of injected labeled cells by dye dilution. T cells were labeled with Cell Trace Violet (CTV), activated *in vitro* with anti-CD3 and anti-CD28 and measured for CTV intensity by flow cytometry at different time-points. Cells did not divide yet at 24 h. The first division can be seen at 36 h and up to 5 divisions can be seen at 72 h. By adoptive transfer of dye-labeled cells, their proliferation can be assessed at later time-points *in vivo*. (**B**) Following the number of cells in the S phase by BrdU or EdU injection. A pulse of nucleoside analogue *in vitro* or *in vivo* labels the cells that are incorporating new DNA in the S phase (replication). An example is given of two cells that perform the cell-cycle phases at different time-points compared to the pulse. The cell in S phase during the pulse, gets a fraction of its DNA labeled depending on its S-phase duration and the pulse duration, while the cell in G1 phase did not get labeled. At the population level, the percent of labeled cells informs on the fraction of cells that were in the S phase during the effective pulse duration, while the percent of labeled DNA inside labeled cells indirectly informs on their S phase duration. (**C**) Tracking of labeled cells at later time-points. A nucleoside analogue pulse (EdU or BrdU) can be followed by tracking the cell-cycle status at different time-points later, informing on the fate of cells that were in S phase during the pulse. (top): six populations can be quantified at each time-point: labeled and unlabeled, and in G0/G1, S or G2/M phases. (bottom) cell-cycle state (% of labeled cells in G0/G1 or S) of whole thymocytes over time after *in vivo* BrdU injection, which already gives an extrapolation of the duration of the G2/M duration (when cells start to be labeled in G1), or the S-phase duration (when labeled cells would have all left the S phase, if they would not come back into G1, by linear extrapolation). The duration of the full cycle, proposed to be when the labeled cells return to S phase, is less straightforward to identify and would need proper mathematical modeling. (**D**) Dual-pulse labeling with EdU followed by BrdU to label cells that enter or leave the S phase in between pulses, and to later track the cycle stage of the labeled cells. (left) scheme of cells that will be labeled by either or nucleoside analogues depending on their cycle stage during the two pulses. (right) Example of visualization of the labeling by flow cytometry at a later time-point, where the DNA level can also be quantified for each population, giving a more precise glance in which stage of the S phase they currently are.

**Figure 5 entropy-23-00437-f005:**
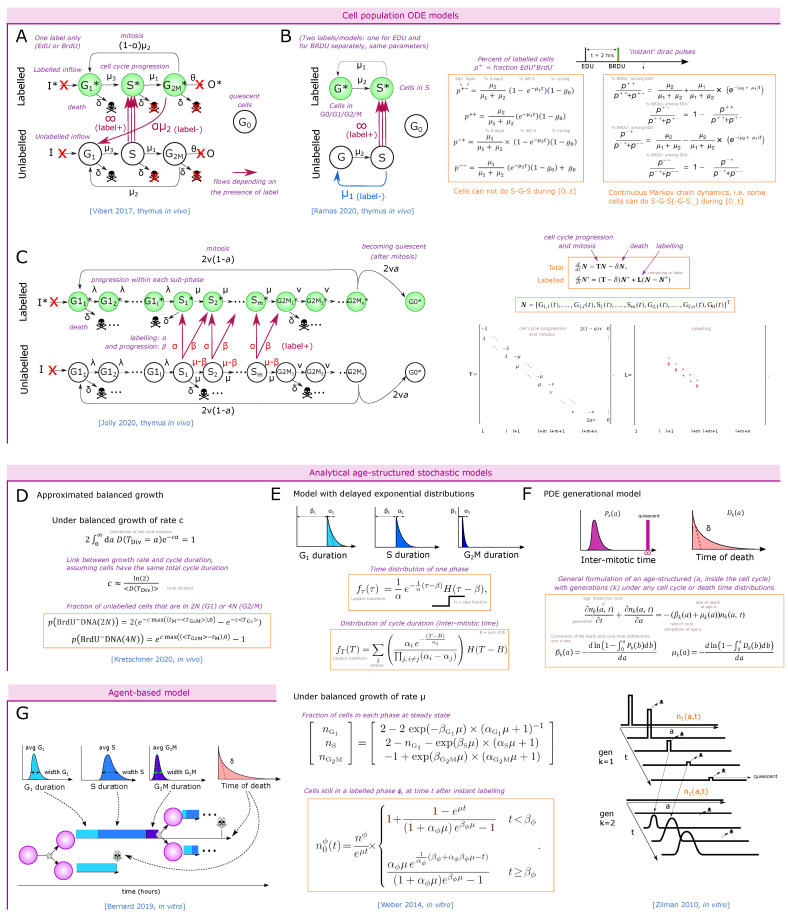
Mathematical approaches used to infer proliferation speed. (**A**–**C**) ODE-based models for simulating *in vivo* labeling of cells. Such models typically model an instant labeling of all cells in S phase, and possibly a decay of the labeling by proliferation (in (**A**) only). In (**B**), a two-pulse labeling is applied, and the dynamics of labeling are simulated for both labels. Assuming instant labeling of all cells in S phase, the first labeling stains the equilibrium value of such cells. Two strategies lead to different analytical formula: assuming the labeling interval *t* is negligible compared to the cell-cycle, cells cannot return in S; or simulating a 2-states Markov chain for the state of the cells at second labeling allows some cells to cycle multiple times. In (**C**), the ODEs can be represented with a matrix formalism. (**D**) From mean-field equations of growing populations, assuming a certain synchrony of the total cycle, the state of initially labeled cells over time can be predicted. (**E**,**F**) Age-structured stochastic models for cell proliferation with time distribution of each cycle phase under exponential growth, assuming delayed exponential distributions (**E**) or with generic cycle and death times convenient when using gamma distributions (**F**). (**G**) Agent-based explicit simulation of each event at the cellular level, pre-defined from time distributions.

**Figure 6 entropy-23-00437-f006:**
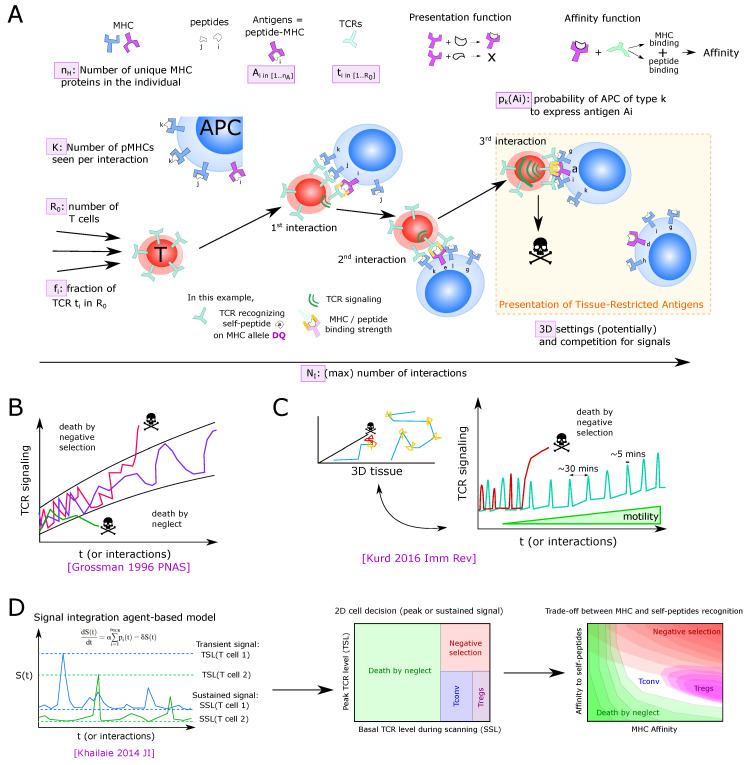
Different biological scales underlying thymic selection and models linking cellular interactions to signal and fate. (**A**) TCR signaling, and thereby thymic selection fate, is mediated by the encounter with Antigen-Presenting Cells (APCs) displaying samples of self-peptides on their MHCs. TCR signaling can be induced by high affinity to an MHC (typically at each interaction), or to a cognate peptide (more rarely). Specific types of APCs express a larger scale of self-antigens (Tissue Restricted Antigens) and are compartmentalized in space (yellow box). (**B**) Model predicting that T cells would show increasing signal over time due to increased TCR expression, and suggesting two self-adapting thresholds, for positive and negative selections. (**C**) Experimental observations on *ex vivo* thymic slices, where T cells migrate and get signaling at each APC encounter. The encounter with cognate peptide leads to stop and strong signaling, while non-self-reactive interactions are shorter. (**D**). Signal integration model. Each encounter with APCs leads to a transient increase in the integrated TCR signaling depending on the affinity (or avidity) of TCR-pMHC binding at each cell interaction. The integrated signal is translated into peak signal (Transient Signaling Level, TSL) and basal signal (Sustained Signaling Level, SSL), used by the T cells to decide their fate. Due to the correlation of SSL with MHC affinity and TSL with highest self-peptide affinity, the decision translates into Tconv with intermediate affinity to MHC while Tregs emerge with higher MHC affinity.

**Figure 7 entropy-23-00437-f007:**
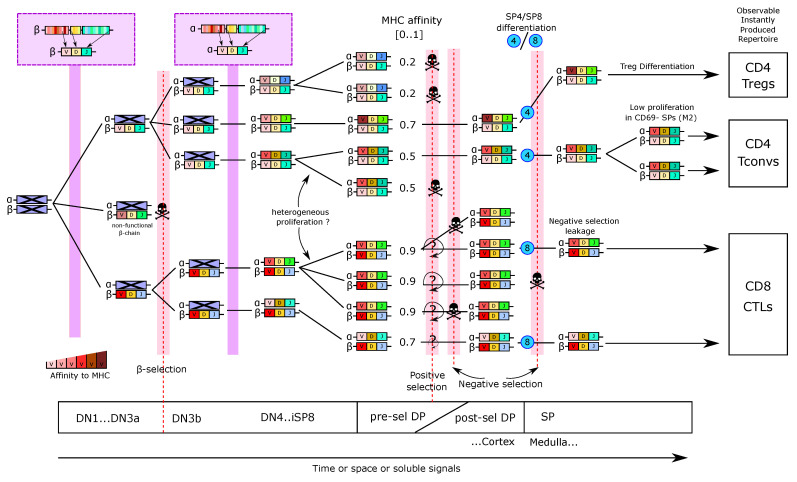
Crosstalk between recombination probabilities, proliferation and selection on the observed TCR frequencies in the repertoire. The progeny of one DN cell is shown as an example, starting from both non-recombined α and β loci (crossed boxes on the left). Recombination events are shown in purple, and each V, D or J fragment is shown with different levels of red, yellow and blue/turquoise. In particular, since the V fragment is responsible for most of the interaction to the MHC, we have colored them to reflect their affinity towards at least one MHC. Due to proliferation in the DN stage, multiple daughter cells carrying different Tcrb gene recombination are ‘tested’ through β selection, proliferate and recombine their Tcra, leading to multiple cells with the same Tcrb recombination but different Tcra recombinations, at the pre-selection DP stage. It is not clear whether cells proliferate equally between Tcrb recombination and the onset of selection. In this example, two cells proliferate once, two cells proliferate twice, and one cell leads to three daughters, as an example. Since the V gene is determining most of the contact interface to the MHC, we have shown an example where only TCRs with high affinity to an MHC survive positive selection (although we have discussed above that this could also be mediated by low affinity peptides). It is not clear whether TCR signaling impacts on the number of divisions, in which case cells with higher MHC affinity (or cross-reactive to multiple low affinity antigens) could proliferate more, which is annotated as a round arrow with a ‘?’. Finally, some cells die by negative selection and differentiate into CD4 Tconvs, CD4 Tregs or CD8 cytotoxic T cells. We have also included low observed proliferation at the SP stage. Altogether, this suggests that the frequencies of TCRs with a recombination scenario can be strongly modified by proliferation and selection between their rearrangement and thymic egress, asking for further mathematical investigation.

## Data Availability

Not applicable.

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
