# Peer review of "Modeling the Dynamics of T-Cell Development in the Thymus"

_entropy, 2021, doi:10.3390/e23040437_

Round 1

Reviewer 1 Report

Robert et al have reviewed the advancement of model and parameter estimations of T cell development in the thymus by compiling more than a hundred of the latest and classical but pivotal papers.

I really appreciate their effort to make such a comprehensive review on this topic.

Because thymic development of T cells in thymus is one of the most quantitatively characterized processes in immunology, this review benefits not only the limited people who are working on thymocyte development but also a broad range of researchers working on population dynamics of immune cells as well as those of virus in tissues and a whole body.

Because I am not an immunologist, I give comments mainly on modelling part below.

I have a couple of questions that I cannot figure out from the review and suggestions to improve the readability of this review.

[Major comments and suggestions]

  • Figure1: Explanation of the right panel in (B) seems to be missing. It should be provided.
  • Figure2: This figure very nicely summarizes different models proposed in the literature. Especially, the labels of papers like [Cai 2007] are very helpful. But the labels are provided only for a subset of the diagrams in fig2 (B). Please provide a complete set of labels in the figure. Also in the middle panel of (A) (the label A is also missing), you may supplement the title of the panel like “ODE with carrying capacity”
  • Line 437: “The capacity of generational models like the one established by Thomas-Vaslin to reproduce fast recovery, would argue that logistic growth is not required. …..”.  I do not understand what the authors want to mean here. Also, I cannot agree with "Generational models can replace logistic growth mechanisms, suggesting that controlled number of divisions is a potential homeostatic mechanism for a fast thymic reconstitution” at Line 1030. I understand that generational models and logistic models describe different aspects of population dynamics. Specifically, the generational models explicitly describe the duration of cell cycle by multiple steps whereas the logistic model represents that the proliferation rate is regulated by the density of cells around a cell. If supra-physiologically high proliferation rate is estimated by a logistic model, I expect that the same result appears in generational models. Also, we can include capacity effect in the generational model by assuming that the speed of cell cycle progression or transition probability of each step in the generational model is inhibited or more generally modulated by the population size of the cells and cellularity in the thymus. Please add more explanations about this matter.
  • Figure 3: This figure very nicely summarizes various experimental methods. But the caption is too blunt and the contents of Fig.3 are not sufficiently explained even in the main body. Please give more explanation about the panels in the figure in caption.
  • Line 554 and others: The authors use “tedious”  a couple of times when mentioning about interpretations of experimental data.  I think this choice of the word not appropriate. Maybe it would be replaced with “complicated” or “tangled” or so on.
  • Figure 4: This figure contains a lot of maths and symbols that are not explained in the review explicitly. I strongly suggest the authors include an appendix to outline the equations and the meaning of symbols in this figure.  For example, we can find a polygonal line in the orange square of panel E. This may means that the function H is a step function. But such implicit assumptions make this figure inaccessible by people who do not have sufficient knowledge about the maths appearing here.
  • Figure 4: The label of panel C is missing. It would be also better if you could put reference labels to the equations in Fig. 4 as you did in Fig. 2.
  • Figure 4 (e,f) and the corresponding part of the main body: The authors use “stochastic models” to indicate the models described in (e) and (f). I think that this terminology is misleading. We and the most theoreticians typically use “stochastic model” to mean a stochastic Markov model described by a Langevin equation or a Master equation.  However, the crucial factor that differentiates the models in (e,f) is explicitly accounting the waiting time distributions between transitions of different states or initiation of the next cell cycle, which are not Markov. We typically use the age-structured model or age-structured PDE model to specify such a stochastic model. I request the authors to use a more specific word than “stochastic model”.
  • Figure 6: Even with the rich content of this figure, the caption is too blunt. Please add more explanation in the caption.
  • Various numerical values appear in this review, but most of them are embedded in the text and are not accessible easily if we want to know that. As one of the authors did in Table 1 of ref [13], could you give a table summarizing parameter values appeared in the references mentioned in this review?
  • The parameter values touched on in this review are from both relatively new results and classical ones. For example, [39] at Line 203 is 30 years old. Can the authors as the specialists of this topic add their opinion about the reliability of the numeric values obtained by such a classical paper? I personally think that such old but important results should be constantly updated or verified by new technologies and measurements. Are there such follow-up results?
  • In the outlook, the authors mainly discussed potential issues and problems about how to model and estimate parameters of thymic T cells. I agree that such problems are important. But it would make this reviewer better if the authors could show what kind of factors we should consider and include in the models from the viewpoint of immunology. For example, in relation to the thymic selections, the cells in thymus other than T cells are important. e.g, thymic B cells, TECs, dendritic cells, and so on. What kind of things should modellers consider with immunologists apart from the model estimation problems.

[Minor questions and comments]

Line 20: "In the absence of an influx of T-lineage competent progenitors, T-cell development may be sustained for extended periods of time [2,3]”.

This sentence looks a bit misleading to beginner because T cell development sounds like being sustained even in a mice that innately lacks the influx. My understanding is that T-cell development can be maintained by the self-replication of thymocytes that already residues in the thymus. If my understanding is correct, It would be better to add a bit more about this sentence

Line 55: I cannot figure out what the authors mean by “spatial”. Is that a spatial difference in the receptor distribution on the membrane? Or is other spatial heterogeneity of intracellular signalling molecules considered?

Line 83: Why the number of niches is 160? Is that a very precise value? Or is there any variation of that?

Line 115: I understand that Porritt et al used purification strategy. Then what should be cared? Should we care about a potential contribution from the omitted progenitor subsets? It would be nice if you could explain a bit more about it.

Line 174: Cai et al [29] are inserted somehow weirdly. Is “and” missing between Manesso and Cai?

Line 277, 414, 432 and others: The word “cellularity” is technical and difficult for beginners to understand its meaning. I guess that the authors use it to mean phenotypes of cells and their population fractions in the thymus but they may mean more. It would be better to provide an explanation about the meaning. 

Line 621: What do you mean by “continuous Markov chain model”? Is it “continuous-time Markov chain model”? Or “Continuous state Markov chain model”? The latter is not so common, though.

Line 704: Why do the authors of that paper (not the authors of this review) insist on log-normal distribution? Does the log-normal distribution fit the experimental data much better than the gamma distribution? In bacterial physiology, we often use gamma distribution because it can fit the experimental cell cycle duration as good as log-normal distribution. I understand that gamma distribution is better when we work on Laplace transform but it would be nice if the authors could add information about the goodness of both distributions for fitting the actual experimental data.

Line 750: What does "both positive and selection combined” mean? Is “negative” missing?

Line 984: You may refer the following attempt: https://www.pnas.org/content/111/27/9875.short

Reviewer 2 Report

The review article by Robert et al. presents an excellent compromise between reviewing the relevant biological processes involved in thymic T cell development and discussing the mathematical modelling strategies for studying those processes. The figures are comprehensive yet clear, offering an overview of key methods and concepts at a glance. This is definitely a valuable contribution to the field, offering to the readers of Entropy a gentle yet broad entry into the mathematical models, their usage, trade-offs, assumptions and limitations to describe T cell population dynamics and differentiation.

Major points:

While the review gives an outstanding account of T cell dynamics at the population level, there is a general lack of insight into relevant intracellular processes. Although briefly mentioned in the section of multiscale approaches, there was little reference to two major components of T cell differentiation and expression dynamics: Gene Regulatory Networks and Stochastic Gene Expression. Expanding on the role of transcriptional regulation and non-coding RNAs on T cell fate decisions, or at least on the necessity of modelling them, would be beneficial to the scope of the review. Likewise, stochastic gene expression of TCRs is likely to play a major role on TCR signalling, being shown and modelled to yield a lognormal distribution of protein expression level, which can vary for individual cells in time (see Guzella TS et al. 2020 PLoS Computational Biology, 16(8), e1007910. DOI: 10.1371/journal.pcbi.1007910). Furthermore, the cell division times and differentiation may be influenced by the circadian clock at the single-cell level, which has been shown to regulate cell cycle progression. These points are not a critical shortcoming of the review that focus on intrathymic T cell development, where in vivo knowledge of intracellular processes is scarce, but they deserve to be mentioned and expanded - at least in the final section.

Finally, the authors could be more explicit in the text about methods at times, especially the ones used for parameter estimation, or the mathematical formalism adopted (e.g. by Grosmann et al. 1996). However, it is important to emphasize that overall the choice of the level of description turned out very clear, yielding a critical but good-flowing narrative. This is important to make the review accessible to an audience that has no expertise in mathematical models.

Minor points:

- Figure 1 & 2 (also applicable to Figs. 4 and 5) legend defining symbols like the hourglass, door and skull (obvious but necessary)

- Figure 4 close bracket on panel B

- pg 6 ln 162 “see section” missing section reference

- pg 8 ln 281 “Trypanosoma cruzi” should be italicized

- pg 13 ln 499 “quantitative training dataset” seems more biased towards machine learning approaches, one could more generally say “quantitative dataset to estimate parameters”

- pg 13 ln 503 repetition of “Thus,” in two adjacent sentences.

- pg 14 ln 520 period missing

- pg 17 ln 628 remove parenthesis from the large (and relevant) sentence

- pg 17 ln 631 remove extra parenthesis closure “)”

- pg 19 ln 714 remove loose parenthesis opening “(”

- Fig 5A close bracket on TCR “ti in [1..R0”

- pg 23 ln 836 sentence could end in a period

- pg 23 ln 858 “mins” and “minutes” used, please replace with “min”

- pg 27 ln 1015 misspelling “moleculatr”

Reviewer 3 Report

The authors present a review that aims to cover the mathematical models that have been used to describe several aspects of thymopoiesis from the number of cell divisions at each development stage, to cell cycle duration, ‘dwell time’ at each differentiation step, efficiency of thymic selection and beyond. They highlight several experimental systems that feed data into these models and the current limitations in these approaches. The authors provide a unique perspective that I have not found covered in the existing literature.

Overall, the authors’ goal is very interesting, but it is perhaps a bit overambitious. I would be in favor of a revised manuscript that is a bit more streamlined in nature such that the reader appreciates the complexities of the models presented - their pros and cons - without being overwhelmed by information.

I highlight below several broad suggestions and provide selected examples; this is not an exhaustive list.

-Improved organization may help streamline and focus the different sections.

More methodical summaries and critiques of the various models in each section is encouraged. Some sections do this better than others, but many ultimately lack a summary of the consistencies and discrepancies among models; in other words, what is generally agreed upon and what is still in dispute?

Paragraph structure should be reconsidered throughout to ensure a theme; some paragraphs lack a proper transition/introduction and tend to be a series of facts that are not well-integrated.

-Precision and clarity in the writing is required.

Line 217, “A more recent continuous labelling study showed that most pre-selection DP became post-selection DP within 4 to 5 days (although they might still proliferate and would never reach 100%)” Consider re-wording as “most” pre-selection DP probably do not undergo selection to become post-selection DP cells. I do not understand the phrase in parentheses.

Line 295, “… while late DPs die by (positive or negative) selection.” What is meant by late DPs die by positive selection?

Line 816, “At the physiological level, the outcome of thymic selection is defined by successful  recognition of foreign peptides (antigens) in the context of self-MHC, resulting in T cell activation.” Positive selection requires self-peptides and does not result in ‘T cell activation’ in the conventional sense.

Line 740, “…negative selection in particular is likely to occur over prolonged periods of time…” This could be taken in multiple ways. For example, it could be interpreted to mean that once a cell gets a negatively selecting signal, the process of deletion/thymocytes apoptosis is long (I do not believe this is what the authors intend), and I do not understand how this is relevant to “upon interaction with more than a single type of thymic APC.”

“Basal” and “Peak” signaling are used to describe (I think) TCR signals that accompany positive and negative selection as read out by intracellular calcium levels; the terms are imprecise and not well defined in the text.

Thymidine labelling is referred to as a ‘dye’ – this is not appropriate.

Figure 1, Endogenous RAG expression is fairly rapidly distinguished after positive selection. In the figure, are the Rag-1 levels that are presented based on endogenous expression of the gene or of a Rag-GFP reporter? In addition, I think it would be helpful to more clearly label the ‘y axis’ of the top panel of A. The ‘Tconv’ terminology might be a bit confusing; I assume you are referring to conventional CD4 T cells here though CD8 T cells are also composed of conventional and unconventional subsets. Consider including “CD4” in the Tconv cells.

-Revise the manuscript to remove information not essential to the review.

There is some redundancy in the description of positive and negative selection on pages 2 and 6.

The the ability of SP cells at the later stages of maturation in response to foreign antigen seems out of place in this review – it is not relevant to normal T cell development.

The RAG-GFP model is introduced somewhat at length in the ‘dye dilution’ section (it is not really a dye) but no data / models build off of this. Is this necessary? It would either need to be expanded upon in its own section or summarized in terms of ‘other tools’ available with, for example, thymidine labelling and why/why not these are used/relevant.

Quantification of TREC are mentioned several times; unless the is reference to quantifying thymic resident cells (rather than recent thymic emmigrants), perhaps this can be removed.

There are several references to Treg that are out of place/seemingly unnecesary.

-The manuscript is generally well referenced though a few statements are missing citations. The authors could also consider (if keeping the sections intact):

PMID: 26522985 is very relevant to the frequency of positive and negative selection among thymocytes and contains other useful references.

PMID: 15654342 is the original thymic slice study of calcium signals in cells undergoing positive selection in thymic slices.

PMID: 24927565 describes the kinetics of positive selection as well as the evolution of calcium signals with differentiation from the DP to SP stage; perhaps this reference was intended in lieu of the methods reference 119 (a methods paper)?

-Minor grammatical errors should be corrected.

Round 2

Reviewer 3 Report

The authors have largely addressed my original comments; this manuscript will be useful resource for the scientific community.